# 4-Aminopyridine promotes functional recovery and remyelination in acute peripheral nerve injury

Kuang-Ching Tseng[1,2], Haiyan Li[1,3], Andrew Clark[3,4], Leigh Sundem[3], Michael Zuscik[1,3], Mark Noble[5,*,†] & John Elfar[1,3,†,**]

## Abstract

**Traumatic peripheral nerve damage is a major medical problem without effective treatment options. In repurposing studies on 4-aminopyridine (4-AP), a potassium channel blocker that provides symptomatic relief in some chronic neurological afflictions, we discovered this agent offers significant promise as a small molecule regenerative agent for acute traumatic nerve injury. We found, in a mouse model of sciatic crush injury, that sustained early 4-AP administration increased the speed and extent of behavioral recovery too rapidly to be explained by axonal regeneration. Further studies demonstrated that 4-AP also enhanced recovery of nerve conduction velocity, promoted remyelination, and increased axonal area post-injury. We additionally found that 4-AP treatment enables distinction between incomplete and complete lesions more rapidly than existing approaches, thereby potentially addressing the critical challenge of more effectively distinguishing injured individuals who may require mutually exclusive treatment approaches. Thus, 4-AP singularly provides both a new potential therapy to promote durable recovery and remyelination in acute peripheral nerve injury and a means of identifying lesions in which this therapy would be most likely to be of value.**

**Keywords** 4-aminopyridine; localized delivery; nerve conduction velocity; peripheral nerve injury; remyelination

**Subject Categories** Neuroscience; Regenerative Medicine

## Introduction

One of the most exciting opportunities for efficient development of new medical interventions is by repurposing existing therapeutic agents for novel applications. A particularly promising approach to such repurposing is to discover situations in which known properties of existing agents can be exploited to provide clinically relevant benefits in settings that are qualitatively different than their established use.

We have applied such a repurposing approach to the medical problem of enhancing functional recovery after traumatic peripheral nerve injuries. Such injuries occur in ~3% of all trauma patients (Taylor *et al*, 2008; Asplund *et al*, 2009; Fex Svennigsen & Dahlin, 2013; Sakuma *et al*, 2015) and can cause severe loss of both motor and sensory functions. A significant subset of injuries [representing, e.g. 45% of peripheral nerve injuries in British soldiers injured in Iraq or Afghanistan between 2005 and 2010 (Birch *et al*, 2012)] exhibit eventual spontaneous recovery, while others require surgical intervention if recovery is ever going to occur. Those injuries in which recovery can occur spontaneously, categorized as neurapraxic lesions, are thought to represent instances in which at least some axons retain continuity through the lesion site. In contrast, if all axons are transected, then recovery depends upon surgical interventions that require cutting of the nerve to present clean ends for rejoining as quickly as possible after injury (Campbell, 2008; Niver & Ilyas, 2014; Ljungquist *et al*, 2015). The decision whether to perform a surgical intervention or not poses a challenging situation for any surgeon. If the surgery is performed too early, any potential for spontaneous recovery is lost. On the other hand, categorization of an injury as neurapraxic is equally risky as the validity of such a diagnosis depends on a spontaneous recovery after weeks or months post-injury (e.g. Shah & Jebson, 2008; Bishop & Ring, 2009; Birch *et al*, 2012). If no recovery occurs, the opportunity for prompt surgical intervention has been lost. Thus, the approaches of extended patient observation and surgical intervention present options that are effectively mutually exclusive.

There are two possible ways to address the above challenges: One is to enhance the speed of recovery in neurapraxic lesions, thus shortening the time required for spontaneous recovery to be observed, while a second is to provide prospective methods of identifying lesions in which axonal continuity exists. Enhancing the speed

1 Center for Musculoskeletal Research, University of Rochester Medical Center, Rochester, NY, USA
2 Department of Chemical Engineering, University of Rochester, Rochester, NY, USA
3 Department of Orthopaedics & Rehabilitation, University of Rochester Medical Center, Rochester, NY, USA
4 Department of Biomedical Engineering, University of Rochester, Rochester, NY, USA
5 Department of Biomedical Genetics, University of Rochester Medical Center, Rochester, NY, USA
  *Corresponding author. Tel: +1 585 273 1448; E-mail: mark_noble@urmc.rochester.edu
  **Corresponding author. Tel: +1 585 273 3157; E-mail: openelfar@gmail.com
  †These authors contributed equally to this work

and/or extent of recovery in neurapraxic lesions is by itself an important medical goal that would improve the quality of life for injured individuals while decreasing costs of medical care and also potentially decreasing the secondary consequences that occur when nerve function is lost (e.g. lost wages, worker productivity). Moreover, the present inability to enhance recovery in neurapraxic lesions, or at least to identify such lesions more quickly, compromises the possible implementation of therapies to improve axonal regeneration in cut nerves by delaying the utilization of such therapies.

The importance of enhancing functional recovery after peripheral nerve injury has spurred multiple research efforts on this topic (e.g. Wan *et al*, 2010a,b; Makoukji *et al*, 2012; Fex Svennigsen & Dahlin, 2013; Stassart *et al*, 2013; McLean *et al*, 2014; Nishimoto *et al*, 2015; Tang *et al*, 2015) with one of the most extensively studied and promising approaches being transient electrical stimulation of the nerve. Electrical stimulation was first reported to increase the speed of functional recovery after experimental peripheral nerve injury over 30 years ago and can promote both axonal regeneration and repair of damage to myelin (e.g. Nix & Hopf, 1983; Pockett & Gavin, 1985; Al-Majed *et al*, 2000; Brushart *et al*, 2002; Ahlborn *et al*, 2007; English *et al*, 2007; Geremia *et al*, 2007; Vivo *et al*, 2008; Haastert-Talini *et al*, 2011; Huang *et al*, 2012; Singh *et al*, 2012). Despite the long-standing interest in electrical stimulation as a potential therapy for peripheral nerve injury, however, implementing this approach has proven challenging (see e.g. Sakuma *et al*, 2015) and it remains relatively unused as a therapeutic strategy.

We hypothesized that if the benefits of electrical stimulation are due to promoting nerve conduction, then pharmacological promotion of impulse conduction should also prove beneficial. We investigated this hypothesis by repurposing of 4-aminopyridine (4-AP), a potassium channel blocker that has not previously been investigated in the context of acute application for traumatic nervous system injury, despite over three decades of study as a potential means of restoring neurological function in a variety of chronic afflictions (e.g. Lundh *et al*, 1977, 1979; Jones *et al*, 1983; Stefoski *et al*, 1987; Hayes *et al*, 1994; Polman *et al*, 1994; Segal & Brunnemann, 1998; Wolfe *et al*, 2001; DeForge *et al*, 2004; Grijalva *et al*, 2010; Claassen *et al*, 2013; Kremmyda *et al*, 2013; Jensen *et al*, 2014; Strupp *et al*, 2014).

In contrast to all previous published studies on 4-AP, we now show that 4-AP is a potent small molecule neuroregenerative agent that enhances both the speed and extent of functional recovery following acute peripheral nerve injury, promotes remyelination, and uniquely among experimental therapies, also enables rapid identification of lesions with axonal continuity. We thus provide a new approach to both treatment and diagnosis of peripheral nerve injuries, based on the discovery of unanticipated benefits of 4-AP in such a setting. The fact that 4-AP is already approved for clinical use for other purposes makes this a compelling candidate to consider for potential clinical studies.

# Results

## 4-AP rapidly enhances functional recovery in acute peripheral nerve injury

As restoration of motor function is the primary goal in treatment of peripheral nerve injury, we first examined the effects of 4-AP administration on this outcome. We employed a standard compression model of sciatic nerve injury (e.g. Magill *et al*, 2007; Elfar *et al*, 2008; Savastano *et al*, 2014) so as not to prejudice experiments toward an absolute requirement for axonal regeneration. To test the clinical relevance of this approach, we initiated treatment 24 h post-injury in order to provide a clinically relevant window of therapeutic opportunity. Mice were treated daily (in the afternoon) with a single injection of 10 μg 4-AP (which approximates a relevant human dose calculated by body weight, but is only ~10% of the mouse body surface area equivalent (Reagan-Shaw *et al*, 2008) of the dosage of 20 mg/day used in treating multiple sclerosis (e.g. Dunn & Blight, 2011; Krishnan & Kiernan, 2013). Motor function was assessed with standard sciatic function index (SFI) analysis (Inserra *et al*, 1998), using measurements of total footprint length, toe spread, and intermediate toe spread. To determine whether improvements were dependent on the presence of drug, all measurements were conducted 20–22 h post-treatment, when the estimated biological half-life of 4-AP in rodents would have caused levels to decrease to < 0.01% of initial dosage (Capacio *et al*, 1996) and effects requiring the presence of 4-AP are no longer observable.

We found that once-daily administration of 10 μg 4-AP enhanced the speed of recovery from crush injury (Fig 1A). As early as 3 days post-injury, mice treated daily (beginning 24 h post-injury) already showed a significant > 25% improvement in gait function over vehicle-treated animals. At 5 and 8 days post-injury, 4-AP-treated mice showed statistically significant twofold greater levels of improvement than vehicle-treated controls.

## Daily 4-AP administration enhances recovery of nerve conduction velocity and does not enhance neuropathic pain responses

As recovery of motor function was too rapid to be explained by axonal regeneration (which occurs at a speed of about 1 mm/day), we considered the possibility that recovery was due to restoration of nerve conduction by other means. To examine this possibility, we first investigated the effects of daily 4-AP administration on restoration of nerve conduction velocity (NCV) after sciatic nerve crush. NCV decreases as a consequence of injury, and restoring the speed of impulse conduction to normal levels is a desired outcome of a regenerative therapy.

Daily 4-AP treatment increased both the speed of improvement in NCV after sciatic nerve crush and the total amount of recovery seen over 35 days (Fig 1B). Nerve crush caused a > 80% decrease in NCV at 7 days post-injury, with only partial recovery over 35 days to 55% of pre-injury NCV. Small, but not yet significant, increases in NCV were seen as early as 7 days after initiation of 4-AP treatment. By 21 days post-injury, 4-AP-treated mice recovered 64% of the normal velocity of conduction followed by 73 and 82% of normal NCV at 28 and 35 days post-injury, respectively (all of which improvements were statistically significant as compared with vehicle-treated mice at all time points).

As peripheral nerve damage often causes neuropathic pain syndromes, and the injury model used in our studies is a standard model for causing such syndromes in the laboratory (e.g. Barriere *et al*, 2009), we also examined the effects of 4-AP treatment on pain responses, with a primary concern of determining whether treatment worsened such responses. In these experiments, injured mice treated with 4-AP were examined for thermal hyperalgesia (Hargreaves assay) or mechanical allodynia (von Frey assay) at 3, 5, 8, 11, 14, and 21 days post-injury.

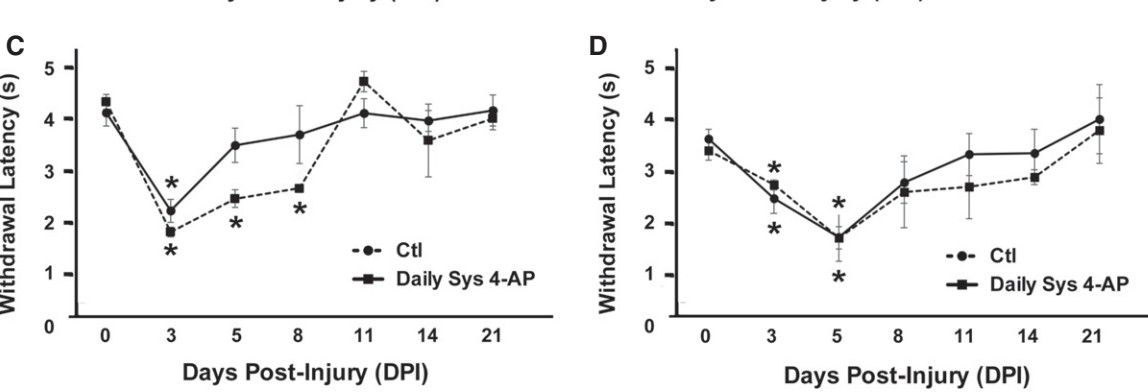

**Figure 1. Daily systemic 4-AP administration (10 µg/day, i.p.) improves functional recovery of crushed sciatic nerve.**

A   Daily 10 µg 4-AP enhanced recovery of sciatic nerve motor function as compared with treatment with vehicle at 3 days post-injury (dpi) through 8 dpi. *: day 3 (D3), *P* = 0.0131; D5, *P* = 0.0475; D8, *P* = 0.0472; ANOVA with *post hoc* comparisons using two-tailed unpaired *t*-test. *n* = 6.

B   Daily 10 µg 4-AP administration also enhanced recovery of nerve conduction velocity (NCV) as observed beginning at 21 days post-injury, eventually restoring NCV to near-normal values while NCV in vehicle-treated mice remained less than half that of uninjured animals. *: D21, *P* = 0.0454; D28, *P* = 0.0487; D35, *P* = 0.0475; ANOVA with *post hoc* comparisons using two-tailed unpaired *t*-test. *n* = 5.

C, D  Analysis of withdrawal latency in response to thermal (C) or mechanical (D) stimuli revealed that 4-AP treatment did not worsen these diagnostics of neuropathic pain syndromes. In the case of response to thermal hyperalgesia, 4-AP-treated mice showed a significantly more rapid return to baseline levels. *: (C) D3, D5, D8: *P* < 0.001 saline versus baseline; D3, *P* = 0.006 4-AP versus baseline; (D) D3, *P* = 0.014; D5, *P* = 0.008; saline versus baseline; D3, *P* = 0.006; D5, *P* = 0.0472; 4-AP versus baseline; ANOVA with *post hoc* comparisons using two-tailed unpaired *t*-test. *n* = 10.

Data information: Data are presented as mean ± SEM and show a representative experiment from 2 to 3 repetitions.

4-AP treatment did not worsen symptoms of neuropathic pain and instead appeared to improve symptoms of thermal hyperalgesia (Fig 1C). Saline-treated injured mice showed significantly increased thermal sensitivity at days 3, 5, and 8 post-injury and returned to baseline sensitivity by day 11, while mice treated with 4-AP only showed increased sensitivity on day 3. Both 4-AP and saline-treated mice showed significantly increased sensitivity to mechanical stimuli at days 3 and 5 post-injury before returning to baseline values, with no differences between the two experimental groups (Fig 1D).

Moreover, there were no changes in response to either type of stimulus in the uninjured limb.

**Localized slow-release administration of 4-AP promotes enhanced recovery**

As one of the common features in studies on electrical stimulation is that each treatment period is limited in length (usually for 30 min, even when administered on multiple days (e.g. Wan *et al*, 2010a,b; McLean *et al*, 2014), it is possible that the pulsatile effects of

transient stimulation are required to obtain benefit. As the short half-life of 4-AP would effectively constitute a pulsatile stimulation, we next utilized localized slow-release formulations of 4-AP to provide continuous administration.

To examine effects of sustained administration of 4-AP, we developed localized slow-release approaches to 4-AP delivery based on encapsulation of 4-AP in poly(lactic-co-glycolic acid) (PLGA, 50:50) microparticles and films, which exhibit different loading capacities and release rates. Loading capacities were 0.8–1.0 µg 4-AP/mg PLGA for particle carriers and 60 µg 4-AP/mg PLGA for films. The releasing profile of (4-AP)-PLGA microparticles *in vitro* was 25.8 µg/mg/day with continued release over ~30 days (Appendix Fig S1). This dosage, released over the course of 24 h, is < 1% of the already low daily dosages delivered by intraperitoneal injections in our first experiments. (4-AP)-PLGA films demonstrated a higher rate of release, with 70% of the 4-AP (i.e. 210 µg) released over the first few days followed by a release rate of ~143 µg/mg/day. We also confirmed that (4-AP)-PLGA carriers labeled with rhodamine and implanted directly onto the crushed nerves (using

PEG hydrogel to hold them in place) remained in place for at least 2 weeks as detected by noninvasive imaging (Appendix Fig S2). Recovery surgery at 3 weeks post-implantation confirmed the continued presence of the PLGA carriers despite the natural motion and inflammatory response to crush injury over the treatment period (Appendix Fig S2). Finally, the bioactivity of 4-AP released from the PLGA delivery systems was confirmed by *in vitro* collection of 4-AP released from PLGA formulations, followed by administration *in vivo*. The 4-AP released *in vitro* after encapsulation appeared to be identical in efficacy to 4-AP that was freshly made up before administration (Appendix Fig S3). Further details on carrier fabrication, loading and analysis are provided in the Appendix.

Sustained release administration of 4-AP was also effective in restoring function after crush injury to the sciatic nerve (Figs 2A and EV1A), even when the total amount of 4-AP delivered was < 0.1% of the total daily dosage applied in our first experiments. In these experiments, either of the two PLGA formulations of 4-AP or identical PLGA formulations not containing 4-AP were placed on the lesion site at the time of injury (so as not to subject mice to a second surgery). As for i.p. administration, we examined motor recovery and NCV recovery.

At 5 days post-injury, mice treated with (4-AP)-PLGA carriers demonstrated significantly improved gait function compared to vehicle-treated controls. Animals implanted with (4-AP)-PLGA films (containing 300 μg 4-AP in 5 mg of PLGA film) had a 40% improvement compared to vehicle-implanted control mice at 3 days post-injury (not statistically different at this time point), while mice implanted with 4-AP-containing particles showed a statistically significant 45% improvement at this time point. Mice implanted with 4-AP-containing films showed a steady improvement in motor function over 14 days, while those implanted with particles showed a brief plateau before an improved recovery. Mice implanted with 4-AP-containing films showed a threefold improvement by day 5 and a 3.5-fold improvement by day 8 post-injury (which was statistically significant for both time points; Fig 2A). Administration of 4-AP in PLGA films showed a trend toward being more effective than delivery in PLGA particles (Fig EV1A).

NCV in mice treated with local 4-AP administration also improved more rapidly than in mice treated with vehicle alone. As with i.p. delivery of 4-AP, the total amount of improvement in NCV in 4-AP-treated mice was greater than that seen in vehicle-treated mice. Significantly greater improvement in NCV was apparent in mice treated with (4-AP)-PLGA particles by the third week post-treatment, indicating that benefit was provided by local activity of 4-AP (as the amount of drug released was so low that systemic effects would not have occurred). Larger improvements were seen in mice treated with (4-AP)-PLGA films (Figs 2B and EV1B; 14.6 ± 1.3 m/s versus 19.2 ± 2.1 m/s versus 27.9 ± 3.2 m/s, respectively, with differences being statistically significant for both Ctl versus particles and Ctl versus films). By 5 weeks post-injury, mice treated with 4-AP-containing films had a 65% faster NCV than vehicle-treated mice (39.2 ± 2.9 m/s versus 25.4 ± 3.6 m/s, a significant difference). As for systemic 4-AP administration, and as expected from patterns of recovery from nerve injury seen clinically, the restoration of near-normal NCV took longer to achieve than restoration of normal SFI.

## Sustained 4-AP treatment enhances neuronal area and myelination and number of myelinated axons

Changes in NCV require changes in axonal structural properties, and the two axonal properties known to contribute to the speed of impulse conduction are cross-sectional area and myelination (e.g. Sanders & Whitteridge, 1946; Waxman, 1980; Ikeda & Oka, 2012 and references therein). We therefore next examined the effects of 4-AP treatment on these parameters.

Injured mice treated with 4-AP films (the most effective of our treatments) showed small but significant increases in average axonal area in nerves examined 21 days after injury and film application (as detected by ultrastructural analysis; Fig 3). This time

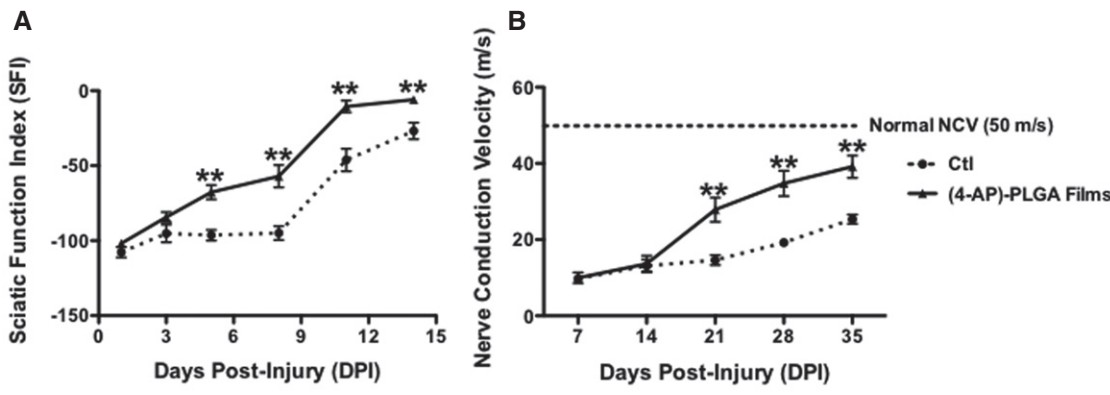

**Figure 2.  Local administration of 4-AP in PLGA films enhances functional and electrophysiological recovery after sciatic nerve crush.**

A   Local 4-AP-treated crushed sciatic nerve (dashed line, vehicle PLGA films; solid line, (4-AP)-PLGA films) regained partial walking ability as early as 3 days post-injury compared to vehicle-treated group. **: D5, *P* = 0.0004; D8, *P* = 0.0012; D11, *P* = 0.003; D14, *P* = 0.0089; ANOVA with *post hoc* comparisons using two-tailed unpaired *t*-test. *n* = 6.

B   Local 4-AP-treated crushed sciatic nerve (dashed line: vehicle PLGA films; solid line: (4-AP)-PLGA films) showed faster improvement in NCV restoration compared with vehicle-treated mice, beginning at 21 days post-injury. **: D21, *P* = 0.0002; D28, *P* = 0.00007; D35, *P* = 0.00019; ANOVA with *post hoc* comparisons using two-tailed unpaired *t*-test. *n* = 5. The figure shows representative outcomes from one of three replicated experiments.

Data information: Data are presented as mean ± SEM and show a representative experiment from 2 to 3 repetitions.

point was examined because this was the earliest time point at which we observed significant NCV changes (Fig 2B). We measured axonal area rather than diameter due to the changes in cross-sectional circularity associated with injury, and we focused attention on myelinated axons because of the role of these fibers in motor function. In uninjured axons, the average area of a myelinated axon (excluding the myelin) was $11.2 \pm 1.3$ μm$^2$. In injured nerves treated with PLGA film only, the average area of myelinated axons decreased to $5.7 \pm 0.2$ μm$^2$ (Fig 3B). In contrast, in nerves treated with 4-AP-containing films, the average area of myelinated axons was $6.7 \pm 0.4$ μm$^2$, a significant improvement versus mice treated only with film. In addition, the proportion of axons with areas above the average value for uninjured nerves was $5 \pm 3\%$ in injured nerves treated with vehicle alone but showed a statistically significant threefold increase to $15 \pm 2\%$ in nerves treated with 4-AP-containing films.

Ultrastructural analysis also revealed that localized 4-AP treatment caused significant changes in myelination post-injury. Mice treated with 4-AP-containing films exhibited an increased myelin thickness and area compared with mice treated with PLGA alone. On day 21 after injury, the average thickness of the myelin within the injured area was $0.53 \pm 0.03$ μm but showed a statistically significant increase to $0.95 \pm 0.14$ μm with localized 4-AP administration, as compared with an average myelin thickness in uninjured nerves of $1.26 \pm 0.08$ μm. In 4-AP-treated mice, $21.4 \pm 10.8\%$ of axons had a myelin thickness above the average for uninjured

nerve, while vehicle-treated mice had no axons with a myelin thickness above the average. We also measured myelin area (Fig 4A) and found the average cross-sectional area of myelin per myelinated large axon in uninjured nerve was $14.8 \pm 1.7$ μm$^2$, was decreased to $4.9 \pm 0.3$ μm$^2$ in injured nerve, and was significantly increased to $10.1 \pm 0.7$ μm$^2$ with localized 4-AP administration. In 4-AP-treated mice, $21.2 \pm 9.3\%$ of axons had myelin areas that were above the average for uninjured nerve, while this value in vehicle-treated mice was only $1.9 \pm 1.4\%$.

Benefits of treatment were also observed by analysis of the ratio of the area of the axon to the area of the axon plus associated myelin. This g$^{area}$-ratio, a variant of the usually employed g-ratio, was calculated due to the decreased circularity in injured nerves. We found that, at 21 days after injury, the average g$^{area}$-ratio in crushed nerves increased significantly from $0.43 \pm 0.01$ (healthy sciatic nerve) to $0.54 \pm 0.06$ (crushed and vehicle-treated sciatic nerve). However, with local 4-AP treatment, the g$^{area}$-ratio improved significantly and was in the normal range of $0.43 \pm 0.09$ (Fig 4B). In addition, in 4-AP-treated mice, $54.4 \pm 18.5\%$ of myelinated axons showed a g$^{area}$-ratio less than the average for uninjured mice, while this value in vehicle-treated mice was $14.4 \pm 1.4\%$.

Increases in myelin were also observed by immunofluorescence and Western blot analysis for the myelin-specific P$_0$ protein. Analysis of tissue lysates at 21 days post-injury/treatment (Fig 4C and D) showed that 4-AP-treated nerves contained $61 \pm 15\%$ more P$_0$ protein in the lesion area than seen in the nerves in vehicle-treated

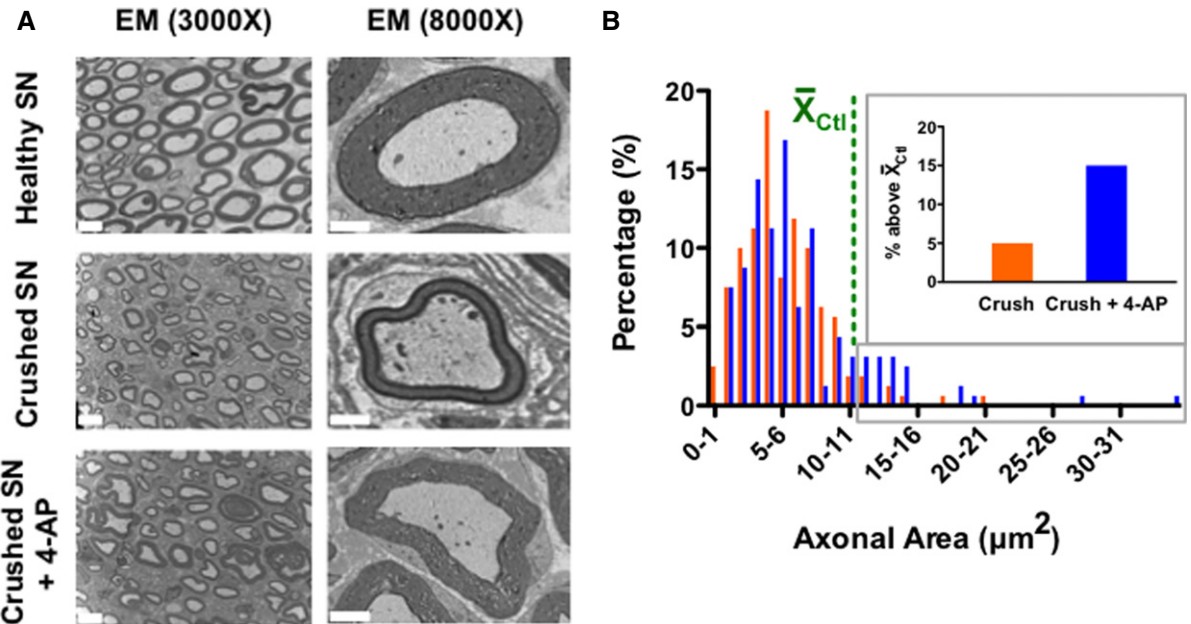

**Figure 3.  Sustained local administration of 4-AP increased axonal area following sciatic nerve crush.**

A   Electron microscopy images of healthy, crushed, and 4-AP-treated crushed sciatic nerve at 21 days post-injury. Scale bar at 3,000× = 5 μm; scale bar at 8,000× = 2 μm.

B   Comparison of the axonal area of randomly chosen individual axons of vehicle-treated and 4-AP-treated mice ($n = 4$ for each experimental group; 40 axons analyzed per mouse). 4-AP-treated sciatic nerve showed statistically greater axonal area compared to the vehicle-treated group ($P < 0.05$; ANOVA; restricted-maximum-likelihood). Moreover, 4-AP-treated mice had a greater proportion of axons with areas greater than the mean value for uninjured mice (shown in the green line), with inset figure displaying all axons with values above this mean.

Data information: This experiment represents a single group of mice of one of three replicates on NCV recovery.

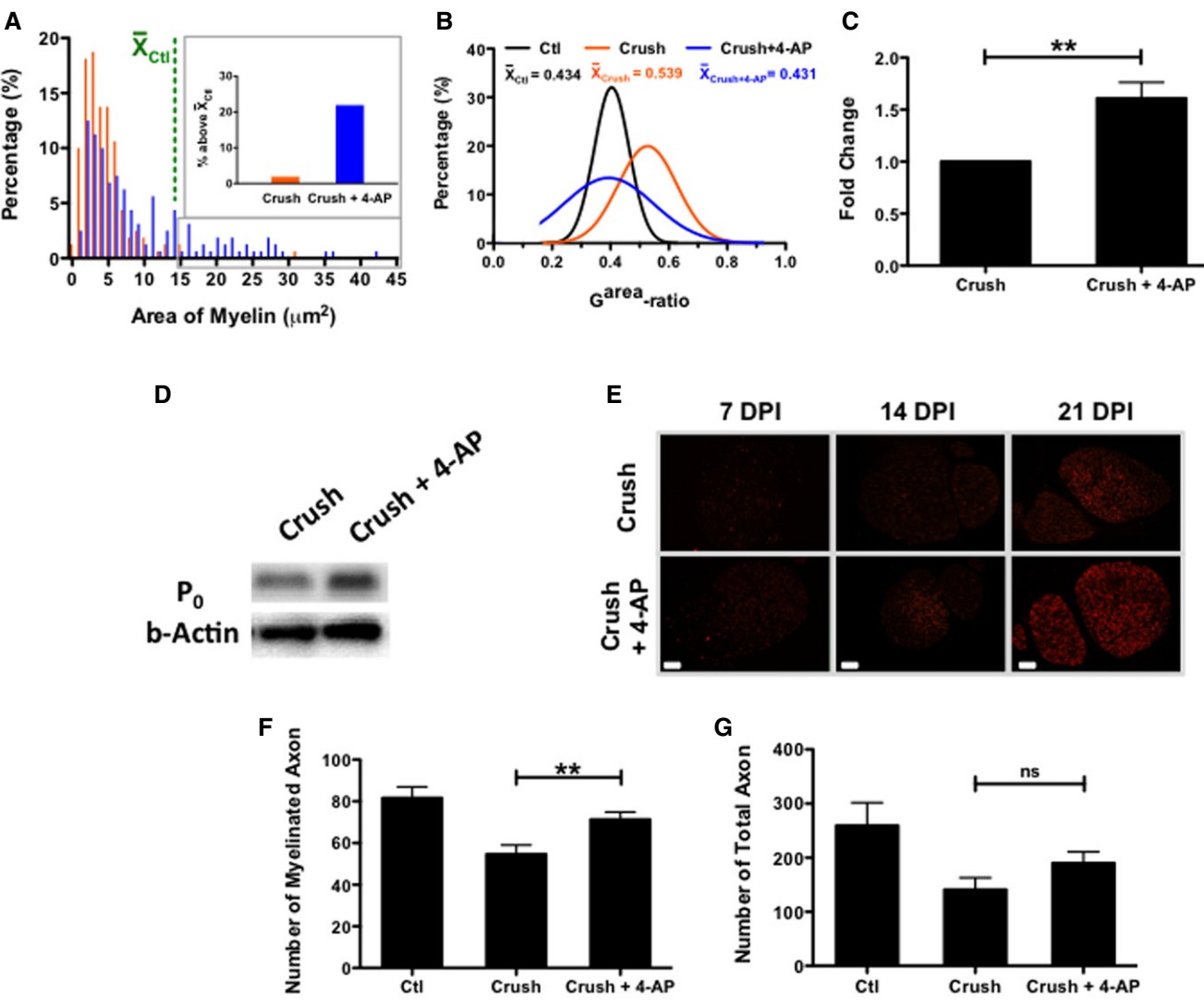

**Figure 4. Long-term/local 4-AP treatment promoted remyelination and increased the number of myelinated axons.**

A, B    Sustained local 4-AP administration was associated with increased myelin area and close-to-normal $g^{area}$-ratio compared to untreated group, as determined by analysis of 40 randomly chosen myelinated axons from sections of 4 nerves for each group ($P < 0.0001$ for saline-treated vs. 4-AP treated nerves; ANOVA with *post hoc* comparisons using two-tailed unpaired *t*-test). 4-AP-treated mice had a greater proportion of axons for which the associated myelin area was greater than the mean value for uninjured mice (shown in the green line), with inset figure displaying all axons with values above this mean.

C, D    4-AP-treated nerves also showed increases in the levels of $P_0$ protein as detected by Western blot analysis (**$P < 0.01$; ANOVA with *post hoc* comparisons using two-tailed unpaired *t*-test).

E       Increases in $P_0$ protein over time also were observed by immunofluorescence analysis at different time points, with $P_0$ protein expression increasing to a greater extent in nerves of mice treated with 4-AP. Scale bars = 200 μm.

F, G    4-AP treatment increased the number of myelinated axons (**$P = 0.002$; ANOVA with *post hoc* comparisons using two-tailed unpaired *t*-test; $n = 4$). Even though the 4-AP-treated group also exhibited a greater number of total axons, this difference was not statistically significant. All myelinated and total axons were counted in five randomly chosen grids from each of 4 nerves for each group. This experiment represents a single group of mice of one of three replicates on NCV recovery.

Data information: Data are presented as mean ± SEM and show a representative experiment from 2 to 3 repetitions.

animals, a significant increase. Levels of $P_0$ protein increased over time as determined by immunofluorescence analysis in sections of crushed nerves treated with vehicle alone or with 4-AP (Fig 4E). The number of myelinated axons was also significantly greater in 4-AP-treated mice examined at 21 days post-injury, as determined by ultrastructural analysis (Fig 4F). The number of total myelinated axons per TEM grid examined (2,310 μm²) was 82 ± 5 axons in

undamaged nerve, as compared with 55 ± 4 axons in injured vehicle-treated mice and 71 ± 4 axons in 4-AP-treated mice, a significant increase over vehicle-treated mice) Similar trends were seen for total axons (Fig 4F; 259 ± 42 versus 140 ± 22 versus 190 ± 21, respectively) and for unmyelinated axons (177 ± 44 versus 86 ± 18 versus 118 ± 19, respectively; not shown), but these differences did not reach statistical significance.

## 4-AP administration in acute injuries enables rapid identification of lesions with axonal continuity

We also examined the possibility that activities of 4-AP could be harnessed to provide a novel solution to the problem of prospectively identifying individuals with axonal continuity. Specifically, in traumatic injuries, it would be valuable to discriminate quickly between

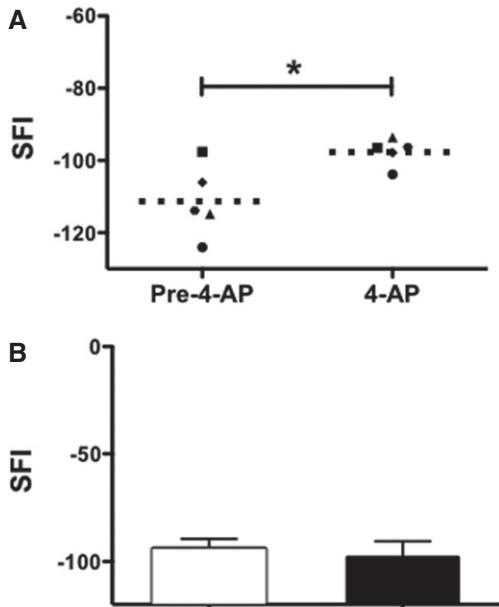

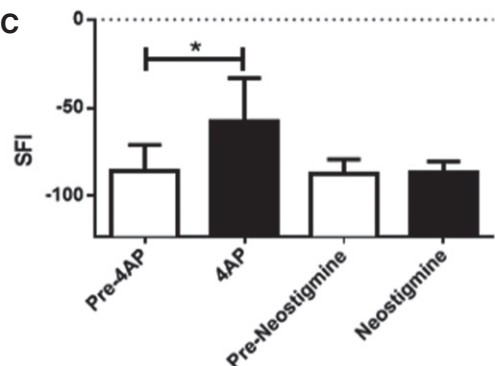

**Figure 5. Systemic 4-AP administration transiently enhances sciatic nerve motor function after crush injury.**

A   At 1 day post-injury, a single dose of 10 μg 4-AP significantly improved walking function as determined by SFI analysis. Horizontal dashed lines represent the mean of each experiment. Each symbol represents a different mouse (*n* = 5; *P = 0.001; ANOVA with *post hoc* comparisons using two-tailed unpaired *t*-test).

B   In contrast, even higher doses of 4-AP administration (50 μg, i.p.) had no effect on SFI in mice with transected nerves (*n* = 10).

C   In contrast to effects of 4-AP, treatment with neostigmine did not cause improvements in SFI (*P = 0.0024 for 4-AP versus saline; 1-way ANOVA with Tukey's multiple comparison test; *n* = 8).

Data information: Data are presented as mean ± SEM and show a representative experiment from 2 to 3 repetitions.

patients in which some axonal continuity exists and those for whom complete axonal transection has occurred because these two groups represent distinct clinical populations that require mutually exclusive interventions. In the former case, preservation and/or enhancement of function of existing axonal connections (e.g. by promoting remyelination) may be able to bring significant benefits. In contrast, in the latter case, benefit can only be achieved if some form of axonal regeneration occurs. Current electro-diagnostic approaches to categorizing nerve injury typically distinguish neurapraxia retrospectively. In the best circumstances, diagnosis can be made 1–6 weeks post-injury, but often requires still longer times (e.g. Robinson, 2000; Lee *et al*, 2004; Perry, 2005; Campbell, 2008; Shah & Jebson, 2008; Bishop & Ring, 2009; Birch, 2010; Birch *et al*, 2012; Kimura, 2013; Ljungquist *et al*, 2015). This delay in diagnosis eliminates opportunities for prompt surgical intervention when such treatment is required.

In comparison with the long periods needed to analyze axonal continuity by electrophysiological approaches, when we examined mice within 1 h of a single injection of 10 μg 4-AP (i.p. 24 h post-injury), we observed significantly improved SFI function in mice with crush injuries, but not in mice with transected nerves (Fig 5). All effects were transient (< 4 h), as predicted from known clearance rates for 4-AP (Uges *et al*, 1982). To examine whether these short-term effects might be the result of 4-AP's ability to enhance synaptic efficacy (e.g. Vizi *et al*, 1977; Lundh, 1978), we also treated mice with neostigmine, a cholinesterase inhibitor that causes increased concentrations of acetylcholine at the neuromuscular junction (e.g. Treffers *et al*, 1988). Neostigmine treatment, however, did not cause any changes in SFI (Fig 5C).

Thus, acute administration of a single treatment with 4-AP enables a rapid distinction to be made between injuries in which the damaged nerve contained axons that traversed the lesion site and those in which all axons were transected.

## Discussion

Sustained administration of 4-AP after acute traumatic peripheral nerve injury offers a novel approach to addressing currently unmet medical needs and provides new uses of this therapeutic agent in settings qualitatively different from its established utility. Despite the lack of prior indications for utility of 4-AP in acute injuries or for enhancing tissue repair, we found that acute initiation of sustained 4-AP treatment enhanced both the speed and extent of recovery of normal motor behavior after crush injury to the sciatic nerve, as analyzed by SFI. Moreover, sustained 4-AP treatment enhanced both the speed and extent of restoration of normal NCV and caused regenerative increases in axonal area, myelin thickness, and levels of the myelin-specific $P_0$ protein. These benefits were not associated with increases in neuropathic pain, as determined by analysis of response to thermal and mechanical stimuli. Indeed, 4-AP treatment appeared to enhance recovery of a normal response to thermal stimuli. Acute transient administration of 4-AP additionally provides a new approach to the identification of nerve injuries in which at least some axons still traverse the lesion site, and thus has additional potential utility as a new approach to diagnosis of peripheral nerve injuries in a manner relevant to initiation of appropriate therapeutic approaches.

Despite being studied for over 30 years in settings of chronic neurological illness, there are no prior indications that 4-AP would

provide durable improvements that are essential for regenerative applications (e.g. Lundh et al, 1977, 1979; Jones et al, 1983; Stefoski et al, 1987; Hayes et al, 1994; Polman et al, 1994; Segal & Brunnemann, 1998; Wolfe et al, 2001; DeForge et al, 2004; Grijalva et al, 2010; Claassen et al, 2013; Kremmyda et al, 2013; Jensen et al, 2014; Strupp et al, 2014). Moreover, the only two published examinations of 4-AP in acute settings provide no evidence for utility in experimental SCI (Haghighi et al, 1995) or in acute vestibular failure (Beck et al, 2014).

We hypothesized that if the widely observed benefits of electrical stimulation in models of peripheral nerve injury are due to simulating nerve conduction, then pharmacologically enabling impulse conduction should also prove beneficial. The only pharmacological agent that is clinically approved for other purposes and that might be useful to achieve this outcome is 4-AP, which enables impulse conduction in demyelinated axons, theoretically by blocking $K^+$ channels that allow leakage of $K^+$ from these axons and thereby enabling axons to restore the level of depolarization required for propagation of action potentials (e.g. Sherratt et al, 1980; Bostock et al, 1981; Targ & Kocsis, 1985; Blight, 1989; Davis et al, 1995; Hayes, 2004).

Several of our findings suggest that 4-AP treatment is an attractive candidate for clinical consideration as a treatment for acute peripheral nerve injury. First, durable and significant improvements in motor function occurred very rapidly (within 3 days post-injury and after just 2 days of treatment), with a speed much greater than could be explained by axonal regeneration. These improvements are unlikely to be due to the presence of residual 4-AP as a single dose of this agent only caused a transient functional improvement, after which SFI returned to pretreatment values. Improvements in NCV also occurred more quickly and were greater than seen in vehicle-treated mice. Although improvements in NCV occurred more slowly than behavioral improvements, such an outcome is consistent with clinical observations. Electro-diagnostic improvement may depend on a greater proportion of neurons acting in a particular way than is required to observe changes in motor function. If 4-AP improved motor function by enabling activity of a relatively small number of demyelinated neurons, this may not be observable by electrophysiological analysis.

The observations that the time course of recovery was too rapid to be due to axonal regeneration led to the unexpected discovery that sustained 4-AP treatment promoted remyelination after crush injury. Remyelination was observed by TEM and by analysis of $P_0$ expression, and improvements in NCV were also consistent with remyelination. Improvements in NCV might also be due, at least in part, to increases in axonal area. As axonal area is increased during myelination (Starr et al, 1996), however, it might be that this change in area is primarily due to repair of myelin damage.

Still a further clinically relevant benefit provided by 4-AP was the ability to cause improved motor function even after a single dose of this agent, delivered 24 h after injury, an observation that could have important consequences for classification and treatment of injuries. The ability of 4-AP to so rapidly improve motor function requires the presence of axonal continuity through the lesion site and appears more likely to be due to enabling axonal conduction than increasing synaptic efficacy, as treatment with the cholinesterase inhibitor neostigmine had no effect on motor behavior. Use of 4-AP to identify lesions with such properties offers the possibility of complementing current approaches to lesion diagnosis, which are dependent on retrospective analysis of recovery, with a means of prospectively identifying lesions in which at least some axonal continuity exists. This would have the dual benefits of being able to assign appropriate patients for treatment with remyelination therapies and also could enable more rapid identification of patients for whom recovery is dependent on surgical intervention. Even if a single 4-AP treatment did not provide these outcomes in clinical settings, shortening the time required to observe recovery would also improve the ability to identify lesions that warrant surgical intervention. Such improvements would be of great importance, as current approaches to diagnosis that rely on a combination of electrodiagnosis and observation of lesions over an extensive period can delay clinical management decisions for as long as 4–6 months, with electrodiagnosis at 7 weeks considered to be an early time point for use of electrodiagnosis in radial nerve injury caused by bone fracture, for example (Shah & Jebson, 2008; Bishop & Ring, 2009). As more rapid repair yields superior outcomes over delayed repair (Campbell, 2008; Niver & Ilyas, 2014; Ljungquist et al, 2015), this also would be a useful application of 4-AP in acute injury. The use of 4-AP to identify lesions with axonal continuity more quickly than is currently possible may also offer a valuable complementary diagnostic classification system to the Seddon and Sutherland classifications originally proposed in 1943 and 1978, respectively (e.g. Campbell, 2008).

Thus, 4-AP administration appears to provide a solution to two of the biggest challenges in treating peripheral nerve injury, by enhancing the speed of recovery in neurapraxic lesions and providing a potential means of even more rapidly identifying lesions in which axonal continuity exists. 4-AP is unique in providing an ability to achieve both of these goals with a single agent and thus differs from other attempts to promote remyelination (e.g. Makoukji et al, 2012; Stassart et al, 2013). Moreover, even if it is correct that 4-AP promotes remyelination by pharmacologically causing similar effects as electrical stimulation (Wan et al, 2010a,b; McLean et al, 2014), there are several reasons why 4-AP may offer a more attractive candidate for clinical studies. First, implantation of sustained release formulations of 4-AP provides clear benefit, while repeated application of electrical stimulation may cause adverse outcomes (Gigo-Benato et al, 2010). 4-AP also is already approved for clinical use, and the studies leading to this approval provide extensive information relevant to potential use of 4-AP in new settings. 4-AP also has the advantage of distributing effectively throughout the body and thus could be used to treat injuries in multiple locations with a single intervention, while use of electrical stimulation has to be applied to defined lesion sites. Even when the location of a lesion is established, our discovery that localized slow-release administration of 4-AP is also therapeutically effective may provide advantages over electrical stimulation by virtue of being applicable as a one-time treatment without requiring multiple clinical visits. We do not yet know if 4-AP can, like electrical stimulation, promote axonal regeneration (a topic of future studies), but even if it does not provide this benefit, its use still would promote remyelination and would enable more rapid identification of lesions in which the surgical interventions required to enable axonal regeneration to occur can be conducted.

Finally, it is important to consider that 4-AP provides a potent example of the greater importance of the activity of an agent, as contrasted with the exact mechanism of action, in developing novel therapeutic interventions. Despite over 30 years of clinical and research studies, it is still unclear whether the benefits provided to individuals with multiple sclerosis are due to enabling conduction in

demyelinated axons, enhancing synaptic efficacy, or both. Indeed, it is not yet clear if the concentrations of 4-AP achieved *in vivo* are high enough to bind to the $K^+$ channels thought to be 4-AP's targets (Dunn & Blight, 2011). Yet, 4-AP has proven to be a safe and effective therapy for many individuals with multiple sclerosis (e.g. Dunn & Blight, 2011; Blight *et al*, 2014; Jensen *et al*, 2014). If its use were held back until such mechanism-related questions were satisfactorily solved, patients would be denied such benefits. We suggest the same considerations are likely to apply to the potential use of 4-AP in treating and diagnosing acute traumatic injury to peripheral nerves.

# Materials and Methods

Further details are provided in the Appendix.

### Reagents and antibodies

4-Aminopyridine, rhodamine-B, poly(D,L-lactide-co-glycolide) (50:50, acid terminated, average MW 38,000–54,000) were from Sigma-Aldrich, anti-$P_0$ monoclonal antibodies from Aves Labs Inc.; anti-β-actin antibodies from Santa Cruz Biotechnology; Hydrogel PEGDM was kindly provided by Danielle S.W. Benoit (University of Rochester).

### Mouse model of peripheral nerve injury

All animal experiments described were approved by the University Committee on Animal Resources (IACUC) at the University of Rochester Medical Center. Anesthetized 10-week-old female C57BL6 mice had the sciatic nerve bluntly exposed directly posterior to the femur. Mice randomly then underwent wound closure without manipulation of the nerve (sham-surgery group), or crush injury of 30-s duration (crush-injury group), as described in, for example (Magill *et al*, 2007; Elfar *et al*, 2008). All experiments were repeated at least three times with a minimum of 5 mice (and generally 8 mice) per experimental group.

For systemic application of 4-AP, drug or saline was injected (10 μg once per day, i.p.) for the duration of the experiment. For localized application of 4-AP, 5 mg of (4-AP)-PLGA particles (containing ~10 μg 4-AP) were suspended in 20 μl PEG hydrogel then photopolymerized in a plastic tube mold (0.02 inches in diameter) to form a (4-AP)-PLGA particles/PEG hydrogel ribbon. This ribbon was placed at the crushed site of the injured nerve immediately after the surgery. For (4-AP)-PLGA films, 5 mg of film (containing 300 μg 4-AP) was shredded to yield fragments of ~1 mm × 3 mm size, which were placed at the crushed site of the sciatic nerve in 20 μl PEG hydrogel, followed by wound closure procedure. Fabrication and release kinetics of (4-AP)-PLGA carriers are discussed in the Appendix.

### Sciatic function index (SFI) determined by walking track analysis

Assessment of motor function recovery was performed by calculating the sciatic function index (SFI) (de Medinaceli *et al*, 1982), as described previously (Elfar *et al*, 2008). Briefly, individual footprints were obtained by painting each foot prior to mice walking a 50 cm path down a narrow corridor lined with paper. Gait was measured from the metrics of resulting footprints: (i) toe spread (TS) (first

through fifth toes), (ii) total print length (PL), and (iii) intermediate toe spread (ITS) (second, third, and fourth toes) of both limbs. If motor dysfunction was so severe as to cause overlap of toe prints, footprints were magnified to enable analysis. All three measurements from three clearly inked randomly chosen footprints per trial were taken from the normal (N) and experimental (E) sides, and the SFI was calculated using the following formula: SFI = −38.3((EPL − NPL)/NPL) + 109.5((ETS − NTS)/NTS) + 13.3((EIT − NIT)/NIT) − 8.8, where E is the injured limb and N is the control limb as in previous studies (Gladman *et al*, 2012).

### Nerve conduction velocity and response to noxious stimuli

Nerve conduction studies were performed by electrical stimulation of a nerve and recording the compound muscle action potential (CMAP) from needle electrodes overlying a muscle supplied by that nerve, as in, for example (Gupta & Steward, 2003; Osuchowski *et al*, 2009). The EMG method was performed with subdermal stainless steel needle electrode placed into the hindlimbs (6 V, 0.1 ms, 1 Hz, 5–15 mA). The stimulating electrode was placed in resting muscle on gluteal fold to obtain the first CMAP. Then, the stimulating electrode was moved to popliteal fossa with a 10 mm fixed distance from gluteal fold to get the second record of CMAP. Nerve conduction velocity (NCV) was determined from the latencies of the potentials and the distance between two stimulating positions (10 mm). Mechanical allodynia and thermal hyperalgesia were determined in injured and uninjured limbs by Von Frey and Hargreaves analyses, as described in the Appendix. Uninjured limbs showed no effect of 4-AP treatment.

### Transmission electron microscopy

Sciatic nerves were immersion fixed and processed by standard procedures (see Appendix). Grids were examined using a Hitachi 7650 TEM and photographed using an attached Gatan Erlangshen 11 megapixel digital camera system.

### Immunofluorescence analyses

Experimental and contralateral (uninjured) sciatic nerves from each test group were harvested at specific time points during healing and recovery and fixed in 4% paraformaldehyde (3 h) and embedded in paraffin. Slides were pretreated with 0.01 M citrate buffer (pH 6.0) for antigen retrieval. Nonspecific blocking was performed with 1:20 diluted serum for 30 min. Sequentially sectioned slides were incubated with primary antibody overnight, followed by incubation with a fluorochrome-labeled secondary antibody for 1 h.

### Immunoblotting analyses

The crushed site of the sciatic nerve at 21 days post-injury was collected and lysed in cell extraction buffer (Invitrogen). Samples were resolved on SDS–PAGE gels and transferred to PVDF membranes (PerkinElmer Life Science, Wellesley, MA, USA). After blocking in 5% bovine serum albumin in PBS containing 0.1% Tween-20, membranes were incubated with a primary antibody, followed by incubation with HRP-conjugated secondary antibody (Santa Cruz Biotechnology).

**The paper explained**

**Problem**

Traumatic peripheral nerve damage is a major medical problem without effective treatment options and in which diagnostic approaches have been static for decades. Even in lesions with the potential for spontaneous recovery, functional restoration occurs slowly, thus impacting quality of life for these individuals and increasing the time to identification of individuals in whom prompt surgical intervention is needed if recovery will ever occur.

**Results**

Results provided in this manuscript indicate that 4-aminopyridine (4-AP), a potassium channel blocker long studied in the context of chronic neurological afflictions, offers significant promise as a small molecule regenerative agent following acute traumatic nerve injury. In crush injuries of the mouse sciatic nerve, 4-AP treatment accelerated behavioral and electrophysiological recovery and enhanced remyelination post-injury. In addition, 4-AP treatment enabled distinction between incomplete and complete lesions more rapidly than existing approaches, thus offering the possibility of more effectively distinguishing between injuries that may require distinct therapeutic approaches.

**Impact**

The ability of 4-AP to promote durable recovery and remyelination following acute traumatic nerve injury offers a potentially valuable new use of this agent as a small molecule regenerative agent able to enhance endogenous repair. As constant daily 4-AP administration is already approved to improve chronic walking disability in multiple sclerosis, transient use for regenerative purposes offers a compelling opportunity for future clinical studies. The additional ability of 4-AP to enable rapid distinction between incomplete and complete nerve injuries means this one drug can potentially be used to identify lesions in which short-term treatment with 4-AP to promote durable recovery would be most likely to be beneficial and also to more rapidly identify individuals for whom timely surgical intervention is required to enhance the likelihood of recovery.

**Image analysis**

Images of cross-sectioned sciatic nerve taken by TEM were processed by ImageJ (US National Institutes of Health, Bethesda, Maryland, USA) to determine myelin area, myelin thickness, and $g^{area}$-ratio on myelinated axonal fibers. Axonal area, axonal circularity, and the number of myelinated and total axons were also counted. For analyzing myelin thickness, 15 randomly chosen myelinated axons were analyzed for each mouse, and 6 thicknesses of myelin sheath (at equal degrees of separation around a central point) were measured on each axon. For determining the axonal area, myelin area and $g^{area}$-ratio: 40 randomly chosen axons were analyzed in each mouse. For analyzing number of myelinated and unmyelinated axons, all axons were counted in each TEM image. Immunofluorescence images of $P_0$ expression in cross-sectioned sciatic nerve were analyzed by ImageJ to determine the average fluorescence intensity of axons-associated $P_0$ labeling.

**Expanded View** for this article is available online.

**Acknowledgements**

The authors thank Margot Mayer-Proschel, Chris Proschel and Hartmut Land for comments, and Karen Bentley for her assistance in the Transmission Electron Microscopy Core. The research was supported, in part, by a grant from the National Institutes of Health to JE (NIH K08 AR060164), by the Friends of Nancy Lieberman Fund (MN) and by the New York State Department of Health Spinal Cord Injury Research Program (C030178).

**Author contributions**

K-CT, HL, AC, and LS conducted experiments; K-CT, AC, and LS conducted primary data analysis; K-CT, MN, and JE conducted further data analysis; and K-CT, MZ, MN, and JE wrote the paper.

**Conflict of interest**

The authors declare that they have no conflict of interest.

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
