## [Review Process File · EMBO Molecular Medicine]

4-aminopyridine promotes functional recovery and remyelination in acute peripheral nerve injury

Kuang-Ching Tseng, Haiyan Li, Andrew Clark, Leigh Sundem, Michael Zuscik, Mark Noble, John Elfar

Corresponding authors: Mark Noble & John Elfar, University of Rochester Medical Center

Review timeline:

Submission date:	04 November 2015
Editorial Decision:	22 December 2015
Revision received:	25 August 2016
Accepted:	29 September 2016

Transaction Report:

Editor: Roberto Buccione

1st Editorial Decision

22 December 2015

Thank you for the submission of your manuscript to EMBO Molecular Medicine and apologies for the delay in reaching a decision on your manuscript.

In this case we experienced unusual difficulties in securing three willing and appropriate reviewers. As a further delay cannot be justified I have decided to proceed based on the two available consistent evaluations.

Although the reviewers have slightly diverging views on your manuscript, they appear to find your work interesting. They do however, have a few concerns of a fundamental nature. I will not dwell into much detail, as their comments are comprehensive. I will just highlight a few main points.

Reviewer 1, as you will see, has three main concerns. Firstly, s/he would also like to see sensory tests performed to increase the potential translational relevance of your findings (which is the point here), given the important pain component peripheral nerve trauma. The Reviewer also feels that some degree of mechanistic analysis on the mode of action of 4-AP must be performed. Finally, s/he notes the unsuited statistical treatment of the data and lack of experimental details.

Reviewer 2 would like more details and discussion on drug delivery and would like you to explore the effect on immune cells. Similarly to Reviewer 1, this Reviewer would also like to see more detail on how the drug acts.

We discussed these evaluations and agreed that these points are well taken and perfectly exemplify the type of information we would like to see in a translationally oriented manuscript. To address these points would significantly enhance the impact and translational relevance of your study.

Reviewer 1 also mentions the need to improve statistical analysis and reporting on animal experimentation. These issues are very close to our hearts at EMBO Press and indeed we ask all authors to take direct action on these aspects upon revision with a mandatory checklist (see further below).

In conclusion, while publication of the paper cannot be published at this stage, we would consider a substantially revised submission, with the understanding that the Reviewers' concerns must be addressed with additional experimental data where appropriate and that acceptance of the manuscript will entail a second round of review.

Please note that it is EMBO Molecular Medicine policy to allow a single round of revision only and that, therefore, acceptance or rejection of the manuscript will depend on the completeness of your responses included in the next, final version of the manuscript.

As you might know, EMBO Molecular Medicine has a "scooping protection" policy, whereby similar findings that are published by others during review or revision are not a criterion for rejection. However, I do ask you to get in touch with us after three months if you have not completed your revision, to update us on the status. Please also contact us as soon as possible if similar work is published elsewhere.

Please note that EMBO Molecular Medicine now requires a complete author checklist (<http://embomolmed.embopress.org/authorguide#editorial3>) to be submitted with all revised manuscripts. Provision of the author checklist is mandatory at revision stage; The checklist is designed to enhance and standardize reporting of key information in research papers and to support reanalysis and repetition of experiments by the community. The list covers key information for figure panels and captions and focuses on statistics, the reporting of reagents, animal models and human subject-derived data, as well as guidance to optimise data accessibility.

I also suggest that you carefully adhere to our guidelines for publication in your next version, including presentation of statistical analyses and our new requirements for supplemental data (see also below) to speed up the pre-acceptance process in case of favourable outcome.

I look forward to seeing a revised form of your manuscript as soon as possible.

***** Reviewer's comments *****

Referee #1 (Comments on Novelty/Model System):

Technical Quality: Behavioral tests evaluating motor function and statistics are not adequate.

Novelty: I am surprised no one has tried 4-AP in PNS injury before. Just because they haven't does not make it novel.

Medical impact: The medical impact has the potential to be significant but the lone motor test gives me pause.

Model System: No concerns

Referee #1 (Remarks):

Peripheral nerve trauma is an important clinical problem. The effectiveness of current clinical therapies is dependent in part upon the proper determination as to the completeness of the injury and timely therapeutic intervention. Incomplete lesions will spontaneously recover to some degree and those are likely to be the target of successful therapies. Identifying a lesion as incomplete within the best therapeutic window can also be challenging. One of the most effective therapies being investigated in the laboratory is electrical stimulation which has been shown to promote axonal regeneration and facilitate myelin repair. Based upon this these authors investigated the therapeutic potential of 4-AP, a potassium channel blocker that is current used clinically, as an effective therapy for traumatic peripheral nerve injury. While the studies are clinically important and for the most part well designed there are concerns that must be addresses.

Concerns:

1. While the test for motor function is supported by solid nerve conduction data the behavioral results are not very satisfying. Deficits demonstrating motor impairments would be strengthened if sensory tests were also included. This is important because chronic neuropathic pain is also a serious clinical problem associated with peripheral nerve trauma. This is especially true if 4-AP is going to be used clinically.
2. Statistical tests: ANOVA should have been used instead of a two tailed T-test. Further, unless I missed it, there isn't a discussion on animal numbers or groups.
3. While some may view that understanding mechanism of action may not be important for repurposing a drug that is already approved for clinical use I disagree. If we understand how a drug works then it may be possible to develop a more effective therapy or use a more effective repurposed compound. Therefore, some attempt at investigating mechanism of action is appropriate.

Referee #2 (Comments on Novelty/Model System):

This is an excellent study which reveals a novel eminently translatable drug for PNS repair.

Referee #2 (Remarks):

In this manuscript the authors study the effect of 4-aminopyridine (4-AP) a potassium channel blocker as a novel therapeutic in PNS injury. Using behavioral/functional outcome measures they saw an enhanced speed of recovery from a crush as well as improvement in gait which was improved if application was given locally. Specifically 4-AP was more beneficial if presented locally using PLGA films. Importantly it was seen that there was an increase in myelination. This is an interesting and exciting report as the drug has already been shown to be safe in CNS injury but its use in PNS injury has been less studied and therefore could be a potential novel therapeutic approach and easy to translate to the clinic.

Specific points

- 1) I would be tempted to put more data in the paper and put Fig 1 and 2 together, and perhaps include more supplementary data particularly on the approach to deliver the drug on films. Could this delivery approach translate to the clinic?
- 2) Was any immunocytochemistry made on the Schwann cells in the nerve? Is it known if the drug affects them? Are there more Schwann cells? is there less Schwann cell death? What does the drug do to the cells in vitro? Could some experiment be added to address these points? Although this reviewer is not asking for a specific mechanism of actions some understanding i) how the films integrate in the tissue and ii) and how the other neural/immune cells look around the site may enhance this paper.
- 3) Was there any effect on macrophages? How does the immune respond to delivery options?

1st Revision - authors' response

25 August 2016

Thank you very much for sending us the constructive reviews on our novel discoveries regarding the ability of 4-aminopyridine to promote durable recovery and remyelination following traumatic peripheral nerve injury.

We are grateful to the reviewers for their support for the clinical potential of our discoveries and are delighted that both reviewers recognized the potential of our findings to move rapidly to clinical analysis. This is the focus of the work, and we are delighted the reviewers were responsive to the possibility that this work eventually could provide benefit to individuals for whom we do not currently have effective therapies.

Reviewer 1 stated the studies were “*clinically important and for the most part well designed and that the medical impact has the potential to be significant.*”

Reviewer 2 stated that “*This is an interesting and exciting report as the drug has already been shown to be safe in CNS injury but its use in PNS injury has been less studied and therefore could be a potential novel therapeutic approach and easy to translate to the clinic.*”

We are delighted to confirm that enthusiasm for the potential clinical relevance of our discoveries is

shared by the United States Food and Drug Administration (FDA) who have recently approved our clinical trial on the use of 4AP as an experimental treatment for iatrogenic traumatic nerve injury caused during radial prostatectomy. This trial is discussed as appropriate in our response to the reviewers.

Response to reviewer requests

The reviewers made some very useful suggestions and we hope they will agree that our attempts to address their concerns have further strengthened our studies. We apologize for the delayed response to their constructive comments. Due to staff turnover, we had to train a new team to carry out experiments relevant to the concerns of the reviewers. As it turned out, this was a useful endeavor not just in further confirming the robustness of our discoveries but in also revealing that the description of how to calculate the sciatic function index (SFI, a standard outcomes measure employed in this field) contained an ambiguity that can lead to a failure to score critical changes in motor function. We have amplified the description of the details of SFI analysis in order to remove this ambiguity. In addition, we wanted to be able to include confirmation that a clinical trial has been approved by the FDA, based on a subset of the observations provided in our manuscript.

As *EMBO Molecular Medicine* publishes (online) the questions of the reviewers and responses of the authors, this enables us to consider the reviewers' concerns in a way that is not possible within the space constraints of the manuscript itself. The comments of the reviewers were very thoughtful and constructive, and we have tried to be equally constructive in our response. We originally hoped to do this by referring the reviewers and readers to reviews on 4AP that discussed the many challenges and paradoxes relevant to understanding this agent, and were disappointed that we were unable to find reviews that brought together all the components we think are needed to address the questions raised in a detailed manner. We therefore decided to err in favor of a detailed response, in the hope of facilitating discussion of topics that we hope will be relevant to the increased interest in 4-AP we think is likely to follow the publication of this manuscript.

We hope the detailed response provided will be useful to the reviewers and editors, and also to the broader readership of *EMBO Molecular Medicine*. We address the reviewer questions in an order that we hope will be most effective in providing information helpful in considering their questions.

Clinical relevance of the studies

As improving the lives of people suffering from injury is our primary goal, we will first address questions with clear relevance to the transition to clinical studies and then consider concerns that are scientifically interesting but, as we discuss, do not appear likely to impact on clinical trial design or even on eventual (hopefully) regulatory approval.

Reviewer 1 stated "*While the test for motor function is supported by solid nerve conduction data the behavioral results are not very satisfying. Deficits demonstrating motor impairments would be strengthened if sensory tests were also included. This is important because chronic neuropathic pain is also a serious clinical problem associated with peripheral nerve trauma. This is especially true if 4-AP is going to be used clinically.*"

We agree completely that this is an important issue, and conducted appropriate experiments to address this question. We carried out von Frey analyses (to examine response to mechanical stimuli) and Hargreaves analyses (to examine response to acute thermal stimulation). As now shown in Figure 1, and accompanying text, we found that 4AP administration did not increase response to either of these standard measures of neuropathic pain syndromes. Indeed, mice treated with 4AP showed a significantly more rapid return to baseline in the case of thermal hyperalgesia and a trend towards more rapid improvement in respect to mechanical allodynia. Moreover, there were no changes in response to either type of stimulus in the uninjured limb.

We are grateful to the reviewer for asking for these important, and clinically relevant, analyses to be added.

Reviewer 1 also stated "*While some may view that understanding mechanism of action may not be*

important for repurposing a drug that is already approved for clinical use I disagree. If we understand how a drug works then it may be possible to develop a more effective therapy or use a more effective repurposed compound. Therefore, some attempt at investigating mechanism of action is appropriate.”

An interest in developing a better understanding of how 4AP might be exerting its benefits was also indicated in the comments of Referee 2.

We are of course also interested in better understanding how 4AP provides its benefits, but it is also important to consider how much mechanistic understanding is required to initiate clinical studies and/or to obtain drug approval, particularly when working with a drug that has been well studied. For scientists focused on mechanism-oriented studies at the cellular and molecular level (including co-corresponding author MN), analysis of 4AP provides a very interesting education about the distances that may exist between the mechanistic analyses we pursue in the laboratory and the type of knowledge needed to provide critical benefit to patients. As this is not a new drug, but an existing drug for which we discovered new uses in addressing currently untreatable injuries, there is a great deal of prior information related to trying to understand its mechanism(s) of action.

Enhanced impulse conduction or enhanced synaptic efficacy (historical view)

Comparison of early laboratory studies on 4AP with subsequent clinical studies reveals the central paradox woven through studies on this agent.

Initial laboratory experiments, and many more recent experiments, were conducted using millimolar concentrations of 4AP more than three orders of magnitude greater than clinically relevant concentrations (which earlier were indicated to be in the range of 1-3 micromolar ([1] and references therein, and more recently were found to be 0.243 +/- 0.113 micromolar in humans receiving the therapeutic dose of 10 mg, twice daily, used in treating walking disability in patients with multiple sclerosis [2]). This paradox is relevant to attempts to understand how 4AP provides its benefits.

The ability of 4AP to enhance neurotransmitter release when applied at supra-clinical doses was recognized early in the course of laboratory studies in the late 1970s (e.g., [3-6]). Such findings led to clinical studies, also initiated in the late 1970s, on Lambert-Eaton myasthenic syndrome [7], which demonstrated a transient enhancement of evoked muscle action potentials with single dose application. Similar results were observed in studies on myasthenia gravis [8] with observations of increased neuromuscular transmission.

The first observations that 4AP enabled impulse conduction in demyelinated axons were made in the early 1980s [9-11], again using millimolar concentrations of this compound. Benefits observed in patients with multiple sclerosis and spinal cord injury – with levels of 4AP administration associated with low or sub-micromolar serum concentrations - bolstered the belief that one of the effects of 4AP treatment is to enable conduction in demyelinated axons (e.g., [11-16]).

These two actions of 4AP, enabling conduction in demyelinated axons and enhancing synaptic efficacy, have dominated thinking about the mechanisms by which this compound works, and have been the basis for initiating multiple clinical trials in a variety of chronic conditions (e.g., [7, 8, 11, 14, 15, 17-46]). The fact that 4AP provides clear benefits in a significant proportion of patients with multiple sclerosis (e.g., [32, 39, 41, 43, 47, 48]), and that evidence of benefit often can be observed promptly after initiation of treatment, is considered to be a strong argument that enhancement of conduction in demyelinated axons is an important effect of 4AP treatment [48]. At the same time, the efficacy of 4AP (and the closely related 3,4-diaminopyridine (DAP)) in Lambert-Eaton myasthenic syndrome (e.g., [7, 8, 17, 49-51]) indicates that enhancement of synaptic function is also an important benefit of such agents (as these syndromes are caused by autoimmune reactions against acetylcholine receptors or other nerve terminal proteins rather than damage to myelin).

Enhanced impulse conduction or enhanced synaptic efficacy (our studies)?

The data presented in our manuscript suggest that the more relevant activity of 4AP in explaining our results is the ability to enable nerve activity in demyelinated axons. Crush injuries are a

standard model of peripheral nerve injuries with a demyelinating component, and have been used previously to study means of promoting remyelination (e.g., [52-58]). Consistent with previous studies using this as a model to study repair of myelin damage, we found a marked fall in levels of the myelin protein P₀ in nerves examined 7 days post-injury (Figure 4E).

The likely role of enhanced nerve conduction as being critical in our studies is particularly indicated by the similarity of results obtained with systemic administration, localized ultra-low dose administration and localized high dose administration. In particular, the ability of standard systemic delivery and local application of 4AP in PLGA beads to promote recovery with concentrations of 4AP that are far less than those achieved in patients are difficult to explain by any other means. We observed a release rate of 25.8 ng/mg/day and mice received 5 mg of particles in their implants (i.e. 129 ng/day). If the daily delivery from the 5 mg of microparticles was treated as a single dose, it would only be ~1.3% of the systemic dose of 10 micrograms/day we administered via intraperitoneal administration (which already is much lower than the equivalent human dosage on the basis of body surface area calculations). As the 4AP is released from PLGA particles continuously, however, the total dose delivered per mouse at any moment in time would be still lower and there is no indication in any previous experiments that such concentrations would have any systemic effect. Within the small volume surrounding the PLGA particles, the drug concentration would of course be much higher than this and, as a result of the effects seen, it seems clear the local drug concentration was in a pharmacologically relevant range. But diffusion away from this site would lead to a rapid fall and extremely low systemic levels. As the reported half-life of 4AP in rodents is two hours [59], serum concentrations would not rise appreciably.

Could conduction have the benefits of promoting enhanced and durable recovery and promoting remyelination? Little is known about this in the context of injury, but axonal activity plays an important role in promoting myelination in the central nervous system during development [60, 61], and may also be an important regulator of Schwann cell development [62, 63]. In addition, as noted in the manuscript, electrical stimulation also has been reported to promote remyelination in experimental peripheral nerve injuries (e.g., [64]).

What of the possibility that the other best-documented activity of 4AP, the ability to enhance synaptic efficacy, is relevant to our results? Based on the comparable efficacy of the localized low-dose formulations of 4AP to systemic administration, this seems improbable. It seems unlikely that the locally applied low-dose 4AP is having any systemic effects, even on nearby synapses (as the site of implantation is 15-25 mm away from the point where muscle joins with nerve). Thus, it does not seem likely that the benefits seen from 4AP exposure, in respect to promoting durable recovery and remyelination, are due to enhanced synaptic efficacy.

As there is little reason to believe that enhanced synaptic function would contribute to promoting durable improvement in motor recovery, electrophysiological function and promotion of remyelination, the most likely situation in which enhanced synaptic efficacy might contribute to improved motor function is shortly after drug administration (as in the studies in the last section of our paper (see Figure 5), focused on the question of whether analysis of the short-term response to 4AP could be used to distinguish nerves with axons traversing the lesion site from those with complete transections). In this setting, 4AP causes a rapid functional recovery that is transient in its nature and theoretically could be due to enhanced synaptic efficacy.

To test the possible contribution of enhanced synaptic efficacy to transient improvements in motor function seen after drug administration, we treated mice with neostigmine, an inhibitor of acetylcholinesterase. Neostigmine has been studied for as long as 4AP, has been both compared and combined with 4AP (e.g., [65-69]), and is well-documented to enhance motor function in myasthenic syndromes in which 4AP and its diamino-derivative 3,4-diaminopyridine (DAP) also have been studied (e.g., [8, 21, 26, 70-73]). Neostigmine, 4AP and DAP all can lead to increased levels of acetylcholine at the neuromuscular junction (which is why both agents, and DAP, have been thought to be relevant to the treatment of myasthenic syndromes (e.g., [7, 8, 17, 49-51])). Unlike 4AP, however, neostigmine works by preventing the breakdown of acetylcholine and in this way causes increased activation of cholinergic postsynaptic receptors. Neostigmine has no known effects on axonal impulse conduction.

When we treated mice with neostigmine at a dose of 0.7 mg/kg, the pediatric dose for reversing

nerve blockade (as extrapolated from [74]) on Days 1 or 2 after injury, we found no improvement in sciatic function index, in contrast with the clear beneficial effects of 4AP even on Day 1 after injury (see Figure 5C of our paper). The lack of effect of neostigmine further suggests that enabling impulse conduction in demyelinated axons is more likely to be of relevance even for this particular outcome.

Molecular analysis of 4AP's mechanism: a challenging problem

More challenging than identifying biological activities of 4AP has been analysis of the mechanism of action of 4AP at the molecular level, and previous studies thus far have not yielded insights of clear clinical relevance. The ability of 4AP to block potassium channels has been known since the mid-1970s (e.g., [75, 76]). As currently understood, 4AP, and DAP, are relatively nonselective blockers of members of the Kv family of voltage-gated potassium channels. 4AP has been more extensively studied, and has the lowest IC₅₀ values for Kv1.5, Kv 3.1 and Kv 3.3 [36, 77]. As determined in heterologous expression systems, however, even here the IC₅₀ values are far above clinically relevant concentrations. Moreover, many studies in the laboratory use still higher concentrations of 4AP, as noted earlier.

One way to ask, however, whether some form of K⁺ channel-related activity is important in promoting durable recovery is to ask whether other K⁺ channel blockers have overlapping activities with 4AP. The two agents that have been studied in most detail in this regard are DAP and tetraethylammonium (TEA)

(with the caveat that many studies on DAP and TEA suffer from the same problem as 4AP in utilizing concentrations that exceed clinically relevant concentrations). TEA has been examined in comparison with 4AP since the late 1970s [78], and the two compounds have been compared in multiple situations (as in, e.g., [9, 10, 79-90]). Although they have overlapping activities and targets [80, 83, 89, 90], and both 4AP and TEA can promote conduction and/or prolong action potentials in demyelinated and pre-myelinated axons [9, 10, 81], they do not have identical targets and actions [84-88]. DAP also is a known K⁺ channel blocker [91, 92].

In response to the interests of the reviewers in inclusion of additional attempts to try and understand the mechanism(s) by which 4AP provides its benefits, we also examined the effects of DAP and TEA on promoting durable recovery from sciatic

nerve crush injury. The experimental paradigm used was identical to that employed with 4AP. DAP was applied at the same dose as 4AP, while TEA was applied at the higher dose of 5mg/kg (based on [74], this being a dose high enough to cause cardiovascular effects in dogs (which have an almost identical LD₅₀ as mice), thus pushing dosage levels to acceptable limits). This is only modestly above the dose of TEA (2mg/kg) reported to provide benefit in a 6-hydroxydopamine model of Parkinson's disease [79], and thus is within the concentration in mice known to be

Figure 1. 3,4-diaminopyridine (34DAP) and tetraethylammonium (TEA) provide small, but in some cases significant, improvements in SFI following peripheral nerve crush injury. 3,4-diaminopyridine and TEA were administered daily at dosages provided in the accompanying text. * = 4-AP vs. saline, $p < 0.0001$ for D1, 2, 5 and 8; TEA vs. saline, $p < 0.0001$ for D3, 5 and 8; 34DAP vs. saline, $p < 0.0001$ for D3, $p = 0.0303$ for D5 (2-way ANOVA with Dunnett multiple comparison test). $N = 3$ for all groups on D1 and D14, $N = 3$ for 4AP and 10 for all other groups D3, 5 and 8.

biologically active.

As shown in the accompanying Figure 1, DAP and TEA both promoted some durable recovery from crush injury, although they were less effective than 4AP, in agreement with previous comparative studies (e.g., [82, 90]). At 3 and 5 days post-injury, both DAP and TEA exposure were associated with small but significant improvements in SFI. These improvements were observed at a time point when all drug levels will have fallen below physiologically effective levels, and thus represent durable changes. A small but significant improvement over saline controls was also seen with TEA on Day 8, which again was smaller than that obtained with 4AP treatment. As these agents all share the ability to promote impulse conduction in demyelinated axons (e.g., [9, 10, 93-95]), which according to the literature is caused by their ability to block K⁺ channels, these observations further support the hypothesis that 4AP is acting through this canonical mechanism (at the cellular level) and most likely is acting through K⁺ channels at the molecular level.

Is there reason to invoke other mechanisms, and how would they be studied?

Reviewer 2 also asked: *“What does the drug do to the cells in vitro? Could some experiment be added to address these points? Although this reviewer is not asking for a specific mechanism of actions some understanding i) how the films integrate in the tissue and ii) and how the other neural/immune cells look around the site may enhance this paper.”*

Reviewer 2 also asked several questions about Schwann cells: *“Was any immunocytochemistry made on the Schwann cells in the nerve? Is it known if the drug affects them? Are there more Schwann cells? is there less Schwann cell death?”*

The attempt of the reviewer to try and suggest possible means of obtaining insights that may shed additional light on the effects of 4AP is appreciated, but we respectfully suggest, for multiple reasons, that such experiments seem unlikely to yield helpful insights.

Perhaps the most frustrating aspect of studying 4AP for those interested in molecular mechanisms is that in vitro studies also suffer from the problem of being unable to study clinically relevant exposure levels. In respect to Schwann cells, others have reported effects of 4AP (such as inhibiting proliferation) when it is applied at hundreds of micromolar to millimolar concentrations [96-99]. What these findings mean in the context of the micromolar (or even submicromolar) concentrations that are relevant in vivo is unknown.

While it theoretically would be possible to study effects of 4AP on, for example, Schwann cell myelination of motor neurons or dorsal root ganglion neurons in vitro, any outcomes of such studies would at best be correlative. Based on the difficulty in equating in vitro and ex vivo results with in vivo outcomes thus far, it seems most likely, moreover, that such studies would require significant expenditures of personnel and funds without providing any information relevant to clinical analyses. Therefore, we hope the reviewer will agree that it is better to defer such studies while we complete analyses of 4AP in other injury paradigms in order to better define general principles that need to be studied. We hope that our present paper also will motivate others to join in studies of this problem in novel ways, which is to be encouraged. But the main focus needs to be on completing and publishing the studies that will motivate others to pursue research on the potential therapeutic properties of 4AP in the context of acute traumatic injury, for which enhancement of functional recovery is of the greatest importance.

In respect to effects of 4AP administration on Schwann cells in vivo, the immunocytochemistry that we showed is P₀ expression, which was critical in determining whether 4AP was preventing the initial loss of myelin. Myelin damage in sciatic nerve crush injury is well established and the marked decreases in P₀ levels observed in injured sciatic nerves at 7 days post injury in both treated and untreated mice suggests that if there is preservation at this level in the immediate aftermath of injury, it is likely to be difficult to detect. Such preservation might be sufficient to provide early enhancement of motor recovery, but it is difficult to see how such studies would provide insights that are other than correlative (as elimination of myelinating Schwann cells would by itself disrupt impulse conduction).

Other components of the Schwann cell response, although a subject of our interest, are at least equally challenging to study and also equally unlikely to provide clinically relevant insights. For example, analyzing cell death is challenging due to the rapid clearance of dead cells, so one has to analyze multiple time points in order to interpret the data. Moreover, even if were possible to detect a difference in cell death it would be very difficult to relate this to the restoration of function.

Thus, even if small and early differences in Schwann cell function were observed in the nerves of mice treated with 4AP, determining whether such changes were relevant to the improved motor recovery, electrophysiological recovery or remyelination would be extremely challenging. Moreover, whether or not such changes were observable would have little impact on the transition of 4AP to clinical analysis.

Perhaps the central challenge in studying preservation or/and enhanced recovery of motor function, however, is that there is a great deal of redundancy built into the nervous system, such that restoration/preservation of function in a small proportion of axons may be sufficient to provide significant motor function (which is the primary functional endpoint one wants to achieve to improve the quality of life for injured individuals). The existence of this redundancy of function seems the most likely explanation as to why behavioral recovery is seen before recovery in nerve conduction velocity, as recovery of a small fraction of neurons may be sufficient to improve behavior but would not be seen in analyzing the broader population of neurons that is collectively studied when analyzing nerve conduction velocity. Once such redundancy is acknowledged, this introduces the challenge that interpreting a failure to find statistically significant changes at the cellular level is not straightforward as behavioral improvements may be disproportionate to what is seen at the tissue level. Moreover, there is the question of how one takes such analyses further. Would a 10% increase in levels of P_0 , or a 10% increase in the number of axons with myelin profiles be meaningful in respect to the functional improvements we saw? And how would one know? (In this respect, it is important to note that we did see an ~30% increase in the number of myelinated axons in our TEM studies performed at Day 21, and presented this data in the manuscript. We prefer, however, only to offer this as a benefit achieved but not attempt to interpret this in the context of functional recovery).

In regards to how the films integrate into the tissue, isn't the most salient finding that systemic administration and local administration (at two widely different dosages) caused the same basic outcomes? Integration of the films into tissue, and effects of films, would certainly warrant further examination if the effects were qualitatively different than occurred with systemic administration, or if the local administration caused adverse effects on recovery. But effects on motor recovery and electrophysiological recovery were essentially indistinguishable between systemic and local administration.

Reviewer 2 also asked *“Was there any effect on macrophages? How does the immune respond to delivery options?”*

Consideration of this question relates to the question of Reviewer 2 about how other neural/immune cells look around the site.

We understand the curiosity of the reviewer and share the desire to obtain as much information as possible (in a resource-appropriate manner). But asking about how the immune system responds to other delivery options is a rather open-ended question, and it's not clear (as discussed below) what such analyses could tell us that would alter the transition to clinical trials of a drug that has been previously studied in pre-clinically and clinically relevant situations as extensively as 4AP (albeit only in situations of chronic injury). This is particularly the case as although there have been multiple studies on the effects of 4AP on lymphocytes and natural killer cells, all such studies have used 4AP concentrations in the range of 130 micromolar to several millimolar (e.g., [100-109]). Thus, these studies pose the same problems as in vitro studies on Schwann cells. Moreover, there is no rationale of which we are aware that would justify invoking involvement of such cells in explaining our results, in contrast with enabling nerve conduction in demyelinated axons.

We do note that there are two interesting reasons to be curious about macrophages. First, macrophage function appears to be modulated in part by K^+ channels, and it has been reported that 4AP exposure can modify the function of macrophages and microglia. However, in one of these

studies [110] the concentration of drug utilized was not provided, and in the others the 4AP concentrations were in the millimolar range typical of in vitro studies [111-113]. A second potential reason to be intrigued about macrophage function is that some reports indicate that PLGA particles can be taken up by macrophages (e.g., [114-117]). However, we found identical results with systemic administration, with PLGA microparticles embedded in PEG (which would be expected to decrease macrophage uptake of particles [117]) and in PLGA films. Thus, it is also unlikely that interaction of the PLGA delivery vehicles with macrophages is relevant to the outcomes obtained.

We have thus far not found any effect of 4AP treatment on macrophage (or microglia) numbers or morphology, including in other injuries in which we are studying 4AP. We do not think it appropriate to discuss this work in detail as of yet, as our studies on macrophages/microglia are limited and are being conducted in multiple tissues. Until we have examined other parameters of function of these cells it seems most judicious to be open to the possibility that we have not yet identified the correct changes on which to focus. That said, the outcomes of our studies thus far make it seem unlikely that effects on macrophages are a primary component of 4AP's effects.

Reviewer 2 also wrote "*I would be tempted to put more data in the paper and put Fig 1 and 2 together, and perhaps include more supplementary data particularly on the approach to deliver the drug on films. Could this delivery approach translate to the clinic?*"

We have now included data on 4AP and the fact that its administration did not enhance neuropathic pain responses (and seemed instead to decrease them), along with data on neostigmine, and provide studies on DAP and TEA in this response. We also have added information to the supplementary information on biodegradation and local retention of the slow-release particles in vivo. We prefer to keep Figure 1 and 2 separate due to their utilization of systemic versus local delivery, respectively.

The question of whether the direct application of slow release 4AP at the site of injury might be translatable to the clinic is one of great interest to us and we thank the reviewer for asking for our thoughts on this. PLGA polymers are approved for clinical use (e.g., [118, 119]) and the ability of an ultra-low dose administration of 4AP to promote repair is particularly attractive for further development. Moreover, the ability to deliver 4AP in a manner that might enable local dose escalation while decreasing the risk of seizurogenic side effects is an attractive possibility for further development.

We are continuing to develop this approach to understand situations in which it can best be applied, which most likely will require a focus on lesions in which the site of injury can be precisely identified. One theoretical possibility would be a further stage of the clinical trial that the FDA has recently approved, which will examine recovery of urinary continence and erectile function after radical prostatectomy. These outcomes are routinely caused by the nerve injuries that occur in this operation, and the location of the injuries is well defined. While the first trial will be conducted using systemic delivery of 4AP, success in this effort would provide an excellent injury for the eventual testing of local delivery approaches.

Other comments of the reviewers

Reviewer 1 also made the following comments:

Novelty: I am surprised no one has tried 4-AP in PNS injury before. Just because they haven't does not make it novel.

We also were surprised. Nonetheless, it appears that no one seems previously to have investigated whether 4AP is of therapeutic utility in the context of acute treatment of traumatic injury in any tissue or even to use 4AP as a diagnostic to distinguish between incomplete and complete nerve injuries (despite the early studies on established demyelinated lesions in peripheral nerve that were important in identifying 4AP's ability to enable conduction in demyelinated axons (e.g., [9-11]).

On the question of novelty, surely it is fair to say that if an agent has been studied for as long as 4AP, in so many different circumstances (and Pubmed lists over 6000 papers in response to the search term "4-aminopyridine"), and we have come up with experiments qualitatively different from the many other studies carried out, then by definition the work is novel (in the sense of being

unusual or different). This is particularly the case as we are not studying the utility of 4AP in a chronic condition that has not yet been studied, but instead have discovered a utility not previously probed. While there are multiple chronic conditions still unexplored for which 4AP might be useful, success in such endeavors will not open up a qualitatively new area of activity. In contrast, our discovery that appropriate early use of 4AP may enable this substance to be used as a regenerative agent is likely to open up new avenues of investigation.

Considering the agreed importance of the goals we have addressed, if someone had previously thought of this possibility, surely they would have tried the experiments. All we can offer as a speculative explanation for the prior lack of investigations of this nature is that it has been embedded in the thinking about 4AP that this agent is useful in certain chronic injuries and perhaps that is the only way in which people previously have thought about its use. Publication of this paper seems likely to have a rapid effect in altering such thinking.

Medical impact: The medical impact has the potential to be significant but the lone motor test gives me pause.

We agree entirely and have added tests on sensory function as correctly requested by the reviewer.

2. Statistical tests: ANOVA should have been used instead of a two tailed T-test. Further, unless I missed it, there isn't a discussion on animal numbers or groups.

In brief, these oversights have been fixed. No changes in p values resulted from these changes. In cases in which only two groups were compared, two-tailed tests also were conducted for each time point comparison between control and treated groups. In addition, in accord with the policies of *EMM*, we have included much more detailed statistical information in each figure legend.

In greater detail, our statistical colleagues do view the question of which statistical analyses to use a bit differently. The point of view provided to us was that the ANOVA test compares multiple (greater than two) groups of data for inter and intra-group variation. The t-test compares two groups for statistical significance in the difference between the means. In each of these tests, a particular pre-test analysis of the variation within the entire data-set must be undertaken to ensure that the data are normally distributed and if so, these two tests (ANOVA for multiple group comparisons greater than 2 and t-tests for two groups) are appropriate. If not, then non-parametric versions of these tests are selected. We undertook this analysis to ensure that our readings or measurements in each experiment were indeed normally distributed.

The view of our statistical colleagues is that ANOVA is appropriate when we were comparing results from three groups. An example would be comparison of the results in functional recovery between locally deliverable forms of PLGA-4AP to control. This analysis did reveal the differences between groups was greater than the differences within groups. The ANOVA analysis also typically includes pairwise comparisons between each of the constituent groups (PLGA-4AP beads, PLGA-4AP films, and untreated controls). This is so that the sources of the inter-group variability which contributed to the significance in ANOVA results can be found. This pairwise comparison (using t-tests) is always necessary so that the source of ANOVA significance can be attributed to the biologically relevant comparison. For example, if we would have found significance in the ANOVA but that significance not carry through to the pairwise (t-test comparison) between each treated (PLGA-4AP) and control group, then the ANOVA would lead us incorrectly to assume that the difference was attributable to treatment instead of perhaps a large difference between the type of treatment and not between treatments and control. It is for this reason that we reported the pairwise comparison data as it precisely shows that each local treatment method is significantly different from control. As this is the most clinically relevant parameter, our ANOVA result is less important in practice, because of the biological implications than the individual pairwise comparisons. However, it should be noted that this is the only place where we had three groups to compare and the rest of the experiments and all of the data presented in this work involves the comparison of two groups of data which are normally distributed.

In any case, as stated, indications of statistical significance were unchanged in comparing the two different approaches to statistical analyses. We do thank the reviewer for asking us to more

thoroughly consider our statistical analyses, with outcomes that seem to emphasize that the changes we observe are both functionally meaningful and statistically significant.

3. While some may view that understanding mechanism of action may not be important for repurposing a drug that is already approved for clinical use I disagree. If we understand how a drug works then it may be possible to develop a more effective therapy or use a more effective repurposed compound. Therefore, some attempt at investigating mechanism of action is appropriate.

As noted in the beginning of this response, the FDA found our data to be sufficiently convincing to approve our first clinical trial. Although this approval would seem to render this particular concern moot, we think instead that the reviewer's comments raise extremely interesting questions related to our general understanding of how drugs provide benefits and to the opportunities offered by drug repurposing. While these are questions that deserve a dedicated review (at a minimum), we would like to offer some preliminary thoughts on this important topic.

As summarized in this response, there remains a great deal to understand about 4AP's mechanism(s) of action. Despite over 30 years of analysis of 4AP in both laboratory and clinical settings, there remains considerable uncertainty about how this substance causes its effects when applied at clinically relevant levels (for review, see, for example, [48]). Could it be that there are other K⁺ channels, as yet to be discovered, which are the real targets of 4AP at the concentrations that are clinically relevant? How can we determine which cells are relevant to the effects of 4AP if in vitro studies require drug concentrations 2-3 orders of magnitude greater than are relevant in vivo? How can we dissect mechanisms in vivo if 4AP binds to multiple K⁺ channels and yields effects that make it unlikely that traditional tools of mouse genetics can be used to help understand cellular targets or mechanisms of action? Might it be that channel properties in injury sites in vivo differ in ways yet unknown from those in circumstances studied thus far? Might it be that the interaction of 4AP with K⁺ channels is different in the open and closed states, such that it becomes trapped in the channel when the channel closes, thus effectively causing levels of 4AP to increase (as suggested more than two decades ago in studies on lymphocytes [104])?

Such questions as those above are all scientifically appropriate, but the weight given to them needs to be balanced against the fact that progress towards understanding the precise mechanisms underlying 4AP's benefits has been slower than identifying important clinical conditions in which this compound (or DAP) provide unambiguous benefits. As we have discussed, the concentrations of 4AP originally used to reveal increases in synaptic efficacy or to enable conduction in demyelinated axons were three orders of magnitude greater than the concentrations that appear to cause the same effects in patients with myasthenic syndromes or multiple sclerosis, respectively.

At the same time that there is a discrepancy between concentrations of 4AP used in early laboratory studies and clinically relevant concentrations, there is no denying that the insights gained from studies using millimolar concentrations of 4AP or DAP in a variety of experimental settings correctly predicted the utility of this agent in enabling functional improvement in syndromes in which neurological function is impaired due to myelin damage or due to autoimmune attack against important synaptic proteins. Studies on multiple sclerosis and myasthenic syndromes were initiated due to the findings made on concentrations of these drugs that far exceed clinically relevant concentrations, yet these studies in experimental systems correctly predicted the outcomes of treatment. What is the evidence, then, that obtaining a better understanding at the level that motivates basic research is going to improve our ability to deploy 4AP therapeutically?

4AP is clinically approved for the treatment of multiple sclerosis and DAP is clinically approved for the treatment of Lambert-Eaton myasthenic syndrome and these drugs provide benefit for patients for whom suitable alternatives do not exist. For many people, the use of these drugs is the difference between living life as an invalid or with the level of function that we wish for all individuals. The types of data we currently seek as basic scientists were not needed to achieve these milestones. To the contrary, had there been insistence on fulfilling such requirements, clinical experiments leading to these therapies would have been very difficult to justify. Nor did we require this level of information to arrive at the testable hypotheses motivating our present studies.

What is going to be the critical path by which a decision is made on whether experimental data on a

particular drug merits going forward to clinical evaluation? Clearly, one should not test substances on patients in the absence of adequate knowledge and safety information. But when a new use has been discovered for an existing drug, which has been well studied for its safety, and where clinically relevant benefits are unequivocally demonstrated in an experimental model widely accepted as clinically relevant, isn't there a case to be made for helping such research to move forward?

Studying the history of analysis of 4AP and DAP is a humbling experience for those of us who are focused on the types of cellular and molecular analyses that dominate the mainstream of biomedical research. While such research has yielded many important insights, it is clear that there are also cases in which an insistence upon such knowledge as a "gatekeeper" for moving forward is not in the patients' best interests. If a drug is safe (and 4AP, used at doses higher than those we find to promote recovery from injury, has been unequivocally safe in individuals with, e.g., multiple sclerosis), then perhaps we need to learn how to think about the problem of drug deployment in a different way than underlies our research on basic scientific discoveries. It seems that in order to have the greatest chance of delivering therapies that are so sorely required, we need to better understand both types of contribution to our knowledge and to promote them both so that greater interest from both directions will converge to most effectively enhance our understanding.

As judged by the FDA, the data in this manuscript on 4AP on promoting recovery from peripheral nerve damage is sufficient to warrant clinical studies, but obtaining clinical information in sufficient numbers is going to be a slow process. Publication of the present studies, in contrast, will promote interest in the previously unrecognized potency of 4AP as a small molecule able to promote tissue repair. Such interest will motivate others to join in the effort to better understand, and learn still new ways, of applying this potential therapy.

References

1. Smith, K.J., P.A. Felts, and G.R. John, *Effects of 4-aminopyridine on demyelinated axons, synapses and muscle tension*. Brain, 2000. **123** p. 171-84.
2. Goodman, A.D., et al., *Fampridine-SR in multiple sclerosis: a randomized, double-blind, placebo-controlled, dose-ranging study*. Mult Scler, 2007. **13**(3): p. 357-68.
3. Johns, A., et al., *The potentiating effects of 4-aminopyridine on adrenergic transmission in the rabbit vas deferens*. Eur J Pharmacol, 1976. **38**(1): p. 71-8.
4. Hue, B., et al., *Synaptic transmission in the sixth ganglion of the cockroach: action of 4-aminopyridine*. J Exp Biol, 1976. **65**(3): p. 517-27.
5. Vizi, E.S., J. van Dijk, and F.F. Foldes, *The effect of 4-aminopyridine on acetylcholine release*. J Neural Transm, 1977. **41**(4): p. 265-74.
6. Illes, P. and S. Thesleff, *4-Aminopyridine and evoked transmitter release from motor nerve endings*. Br J Pharmacol, 1978. **64**(4): p. 623-9.
7. Agoston, S., et al., *Effects of 4-aminopyridine in Eaton Lambert Syndrome*. Br J Anaesth, 1978. **50**(4): p. 383-5.
8. Lundh, H., O. Nilsson, and I. Rosen, *Effects of 4-aminopyridine in myasthenia gravis*. J Neurol Neurosurg Psychiatry, 1979. **42**(2): p. 171-5.
9. Sherratt, R.M., H. Bostock, and T.A. Sears, *Effects of 4-aminopyridine on normal and demyelinated mammalian nerve fibres*. Nature, 1980. **283**: p. 570-572.
10. Bostock, H., T.A. Sears, and R.M. Sherratt, *The effects of 4-aminopyridine and tetraethylammonium ions on normal and demyelinated mammalian nerve fibres*. J Physiol, 1981. **313**: p. 301-15.
11. Jones, R.E., et al., *Effects of 4-aminopyridine in patients with multiple sclerosis*. J Neurol Sci, 1983. **60**(3): p. 353-62.
12. Bever, C.J., et al., *The effects of 4-aminopyridine in multiple sclerosis patients: results of a randomized, placebo-controlled, double-blind, concentration-controlled, crossover trial*. Neurology, 1994. **44**: p. 1054-1059.
13. Blight, A.R. and J.A. Gruner, *Augmentation by 4-aminopyridine of vestibulospinal free fall responses in chronic spinal-injured cats*. J. Neurol. Sci., 1987. **82**: p. 145-159.
14. Davis, F.A., D. Stefoski, and J. Rush, *Orally administered 4-aminopyridine improves clinical signs in multiple sclerosis*. Ann Neurol, 1990. **27**(2): p. 186-92.
15. Stefoski, D., et al., *4-Aminopyridine improves clinical signs in multiple sclerosis*. Ann Neurol, 1987. **21**(1): p. 71-7.

16. Van Diemen, H.A., et al., *4-Aminopyridine in patients with multiple sclerosis: dosage and serum level related to efficacy and safety*. Clin Neuropharmacol, 1993. **16**(3): p. 195-204.
17. Lundh, H., O. Nilsson, and I. Rosen, *4-aminopyridine--a new drug tested in the treatment of Eaton-Lambert syndrome*. J Neurol Neurosurg Psychiatry, 1977. **40**(11): p. 1109-12.
18. Blight, A.R., *Effect of 4-aminopyridine on axonal conduction-block in chronic spinal cord injury*. Brain Res. Bull., 1989. **22**: p. 47-52.
19. van Diemen, H.A., et al., *The effect of 4-aminopyridine on clinical signs in multiple sclerosis: a randomized, placebo-controlled, double-blind, cross-over study*. Ann Neurol, 1992. **32**(2): p. 123-30.
20. Hansebout, R.R., et al., *4-Aminopyridine in chronic spinal cord injury: a controlled, double-blind, crossover study in eight patients*. J. Neurotrauma, 1993. **10**: p. 1-18.
21. Sanders, D.B., J.F. Howard, Jr., and J.M. Massey, *3,4-Diaminopyridine in Lambert-Eaton myasthenic syndrome and myasthenia gravis*. Ann N Y Acad Sci, 1993. **681**: p. 588-90.
22. Hayes, K.C., et al., *4-Aminopyridine-sensitive neurologic deficits in patients with spinal cord injury*. J Neurotrauma, 1994. **11**(4): p. 433-46.
23. Polman, C.H., et al., *4-Aminopyridine is superior to 3,4-diaminopyridine in the treatment of patients with multiple sclerosis*. Arch Neurol, 1994. **51**(11): p. 1136-9.
24. Haghghi, S.S., et al., *Effect of 4-aminopyridine in acute spinal cord injury*. Surg Neurol, 1995. **43**(5): p. 443-7.
25. Anlar, B., et al., *3,4-diaminopyridine in childhood myasthenia: double-blind, placebo-controlled trial*. J Child Neurol, 1996. **11**(6): p. 458-61.
26. Molgo, J. and J.M. Guglielmi, *3,4-Diaminopyridine, an orphan drug, in the symptomatic treatment of Lambert-Eaton myasthenic syndrome*. Pflugers Arch, 1996. **431**(6 Suppl 2): p. R295-6.
27. Qiao, J., et al., *Effects of 4-aminopyridine on motor evoked potentials in patients with spinal cord injury*. J. Neurotrauma, 1997. **14**: p. 135-149.
28. Segal, J.L. and S.R. Brunnemann, *4-Aminopyridine improves pulmonary function in quadriplegic humans with longstanding spinal cord injury*. Pharmacotherapy, 1997. **17**(3): p. 415-23.
29. Potter, P.J., et al., *Randomized double-blind crossover trial of fampridine-SR (sustained release 4-aminopyridine) in patients with incomplete spinal cord injury*. J Neurotrauma, 1998. **15**(10): p. 837-49.
30. Segal, J.L. and S.R. Brunnemann, *4-Aminopyridine alters gait characteristics and enhances locomotion in spinal cord injured humans*. J Spinal Cord Med, 1998. **21**(3): p. 200-4.
31. Segal, J., et al., *Safety and efficacy of 4-aminopyridine in humans with spinal cord injury: a long-term, controlled trial*. Pharmacotherapy, 1999. **19**: p. 713-723.
32. Rossini, P.M., et al., *Fatigue in progressive multiple sclerosis: results of a randomized, double-blind, placebo-controlled, crossover trial of oral 4-aminopyridine*. Mult. Scler., 2001. **7**: p. 354-358.
33. van der Bruggen, M.A., et al., *Randomized trial of 4-aminopyridine in patients with chronic incomplete spinal cord injury*. J. Neurol., 2001. **248**: p. 665-671.
34. Wolfe, D.L., et al., *Effects of 4-aminopyridine on motor evoked potentials in patients with spinal cord injury: a double-blinded, placebo-controlled crossover trial*. J. Neurotrauma, 2001. **18**: p. 757-771.
35. Strupp, M., et al., *Aminopyridines for the treatment of cerebellar and ocular motor disorders*. Prog Brain Res, 2008. **171**: p. 535-41.
36. Alvina, K. and K. Khodakhah, *The therapeutic mode of action of 4-aminopyridine in cerebellar ataxia*. J Neurosci, 2010. **30**(21): p. 7258-68.
37. Grijalva, I., et al., *High doses of 4-aminopyridine improve functionality in chronic complete spinal cord injury patients with MRI evidence of cord continuity*. Arch Med Res, 2010. **41**(7): p. 567-75.
38. Kalla, R., et al., *Comparison of 10-mg doses of 4-aminopyridine and 3,4-diaminopyridine for the treatment of downbeat nystagmus*. J Neuroophthalmol, 2011. **31**(4): p. 320-5.
39. Sedehizadeh, S., M. Keogh, and P. Maddison, *The use of aminopyridines in neurological disorders*. Clin Neuropharmacol, 2012. **35**(4): p. 191-200.
40. Claassen, J., et al., *A randomised double-blind, cross-over trial of 4-aminopyridine for downbeat nystagmus--effects on slowphase eye velocity, postural stability, locomotion and symptoms*. J Neurol Neurosurg Psychiatry, 2013. **84**(12): p. 1392-9.
41. Goodman, A.D. and R.T. Stone, *Enhancing neural transmission in multiple sclerosis (4-aminopyridine therapy)*. Neurotherapeutics, 2013. **10**(1): p. 106-10.

42. Kremmyda, O., et al., *4-Aminopyridine suppresses positional nystagmus caused by cerebellar vermis lesion*. J Neurol, 2013. **260**(1): p. 321-3.
43. Jensen, H.B., et al., *4-Aminopyridine for symptomatic treatment of multiple sclerosis: a systematic review*. Ther Adv Neurol Disord, 2014. **7**(2): p. 97-113.
44. Strupp, M., et al., *Pharmacotherapy of vestibular and cerebellar disorders and downbeat nystagmus: translational and back-translational research*. Ann N Y Acad Sci, 2015. **1343**: p. 27-36.
45. Feil, K., et al., *Update on the Pharmacotherapy of Cerebellar Ataxia and Nystagmus*. Cerebellum, 2016. **15**(1): p. 38-42.
46. Kalla, R., et al., *Update on the pharmacotherapy of cerebellar and central vestibular disorders*. J Neurol, 2016. **263 Suppl 1**: p. 24-9.
47. Hayes, K.C., *The use of 4-aminopyridine (fampridine) in demyelinating disorders*. CNS Drug Rev, 2004. **10**(4): p. 295-316.
48. Dunn, J. and A. Blight, *Dalfampridine: a brief review of its mechanism of action and efficacy as a treatment to improve walking in patients with multiple sclerosis*. Curr Med Res Opin, 2011. **27**(7): p. 1415-23.
49. Lundh, H., *Effects of 4-aminopyridine on neuromuscular transmission*. Brain Res, 1978. **153**(2): p. 307-18.
50. Muller, D., *Potentialiation by 4-aminopyridine of quantal acetylcholine release at the Torpedo nerve-electroplaque junction*. J Physiol, 1986. **379**: p. 479-93.
51. Thomsen, R.H. and D.F. Wilson, *Effects of 4-aminopyridine and 3,4-diaminopyridine on transmitter release at the neuromuscular junction*. J Pharmacol Exp Ther, 1983. **227**(1): p. 260-5.
52. Wan, L., R. Xia, and W. Ding, *Short-term low-frequency electrical stimulation enhanced remyelination of injured peripheral nerves by inducing the promyelination effect of brain-derived neurotrophic factor on Schwann cell polarization*. J Neurosci Res, 2010. **88**(12): p. 2578-87.
53. Wan, L.D., R. Xia, and W.L. Ding, *Electrical stimulation enhanced remyelination of injured sciatic nerves by increasing neurotrophins*. Neuroscience, 2010. **169**(3): p. 1029-38.
54. Makoukji, J., et al., *Lithium enhances remyelination of peripheral nerves*. Proc Natl Acad Sci U S A, 2012. **109**(10): p. 3973-8.
55. Stassart, R.M., et al., *A role for Schwann cell-derived neuregulin-1 in remyelination*. Nat Neurosci, 2013. **16**(1): p. 48-54.
56. Fex Svennigsen, A. and L.B. Dahlin, *Repair of the peripheral nerve-Remyelination that works*. Brain Sci, 2013. **3**(3): p. 1182-97.
57. McLean, N.A., et al., *Delayed nerve stimulation promotes axon-protective neurofilament phosphorylation, accelerates immune cell clearance and enhances remyelination in vivo in focally demyelinated nerves*. PLoS One, 2014. **9**(10): p. e110174.
58. Tang, Y.D., et al., *Nimodipine-mediated re-myelination after facial nerve crush injury in rats*. J Clin Neurosci, 2015.
59. Capacio, B.R., et al., *A method for determining 4-aminopyridine in plasma: pharmacokinetics in anaesthetized guinea pigs after intravenous administration*. Biomed Chromatogr, 1996. **10**(3): p. 111-6.
60. Demerens, C., et al., *Induction of myelination in the central nervous system by electrical activity*. Proc Natl Acad Sci U S A, 1996. **93**(18): p. 9887-92.
61. Gibson, E.M., et al., *Neuronal activity promotes oligodendrogenesis and adaptive myelination in the mammalian brain*. Science, 2014. **344**(6183): p. 1252304.
62. Stevens, B., S. Tanner, and R.D. Fields, *Control of myelination by specific patterns of neural impulses*. J Neurosci, 1998. **18**(22): p. 9303-11.
63. Stevens, B. and R.D. Fields, *Response of Schwann cells to action potentials in development*. Science, 2000. **287**(5461): p. 2267-71.
64. Huang, J., et al., *Electrical stimulation to conductive scaffold promotes axonal regeneration and remyelination in a rat model of large nerve defect*. PLoS One, 2012. **7**(6): p. e39526.
65. Treffers, R., A.L. Frankhuyzen, and L.H. Booij, *Effects of neostigmine, edrophonium, 4-aminopyridine and their combinations*. Acta Anaesthesiol Belg, 1988. **39**(1): p. 55-8.
66. Tierney, P.C., Y.I. Kim, and T.R. Johns, *Synergistic interaction of 4-aminopyridine with neostigmine at the neuromuscular junction*. Eur J Pharmacol, 1985. **115**(2-3): p. 241-7.
67. Wirtavuori, K., M. Salmenpera, and T. Tammisto, *Antagonism of d-tubocurarine-induced neuromuscular blockade with a mixture of 4-aminopyridine and neostigmine in man*. Can Anaesth Soc J, 1984. **31**(6): p. 624-30.

68. Miller, R.D., et al., *4-Aminopyridine potentiates neostigmine and pyridostigmine in man*. *Anesthesiology*, 1979. **50**(5): p. 416-20.
69. Uchiyama, T., M. Lemeignan, and P. Lechat, *Reversibility of kanamycin-induced neuromuscular and cardiovascular depressions by 4-aminopyridine, in comparison with neostigmine*. *Jpn J Antibiot*, 1982. **35**(6): p. 1405-10.
70. Lundh, H., O. Nilsson, and I. Rosen, *Treatment of Lambert-Eaton syndrome: 3,4-diaminopyridine and pyridostigmine*. *Neurology*, 1984. **34**(10): p. 1324-30.
71. Lundh, H., O. Nilsson, and I. Rosen, *Improvement in neuromuscular transmission in myasthenia gravis by 3,4-diaminopyridine*. *Eur Arch Psychiatry Neurol Sci*, 1985. **234**(6): p. 374-7.
72. Palace, J., C.M. Wiles, and J. Newsom-Davis, *3,4-Diaminopyridine in the treatment of congenital (hereditary) myasthenia*. *J Neurol Neurosurg Psychiatry*, 1991. **54**(12): p. 1069-72.
73. Maddison, P., *Treatment in Lambert-Eaton myasthenic syndrome*. *Ann N Y Acad Sci*, 2012. **1275**: p. 78-84.
74. Barnes, L.G. and C.D. Eltherington, *Dug Dosage in Laboratory Animals: A Handbook*. 1973, California: University of California Press.
75. Yeh, J.Z., et al., *Interactions of aminopyridines with potassium channels of squid axon membranes*. *Biophys J*, 1976. **16**(1): p. 77-81.
76. Llinas, R., K. Walton, and V. Bohr, *Synaptic transmission in squid giant synapse after potassium conductance blockage with external 3- and 4-aminopyridine*. *Biophys J*, 1976. **16**(1): p. 83-6.
77. Coetzee, W.A., et al., *Molecular diversity of K⁺ channels*. *Ann N Y Acad Sci*, 1999. **868**: p. 233-85.
78. Fink, R. and E. Wettwer, *Modified K-channel gating by exhaustion and the block by internally applied TEA⁺ and 4-aminopyridine in muscle*. *Pflugers Arch*, 1978. **374**(3): p. 289-92.
79. Haghdoost-Yazdi, H., et al., *Significant effects of 4-aminopyridine and tetraethylammonium in the treatment of 6-hydroxydopamine-induced Parkinson's disease*. *Behav Brain Res*, 2011. **223**(1): p. 70-4.
80. Brooke, R.E., et al., *Kv3 voltage-gated potassium channels regulate neurotransmitter release from mouse motor nerve terminals*. *Eur J Neurosci*, 2004. **20**(12): p. 3313-21.
81. Devaux, J., et al., *Effects of K⁺ channel blockers on developing rat myelinated CNS axons: identification of four types of K⁺ channels*. *J Neurophysiol*, 2002. **87**(3): p. 1376-85.
82. Schechter, L.E., *The potassium channel blockers 4-aminopyridine and tetraethylammonium increase the spontaneous basal release of [3H]5-hydroxytryptamine in rat hippocampal slices*. *J Pharmacol Exp Ther*, 1997. **282**(1): p. 262-70.
83. Bethge, E.W., et al., *Effects of some potassium channel blockers on the ionic currents in myelinated nerve*. *Gen Physiol Biophys*, 1991. **10**(3): p. 225-44.
84. Puil, E., R.M. Miura, and I. Spigelman, *Consequences of 4-aminopyridine applications to trigeminal root ganglion neurons*. *J Neurophysiol*, 1989. **62**(3): p. 810-20.
85. Eng, D.L., et al., *Development of 4-AP and TEA sensitivities in mammalian myelinated nerve fibers*. *J Neurophysiol*, 1988. **60**(6): p. 2168-79.
86. Saint, D.A., D.M. Quastel, and Y.Y. Guan, *Multiple potassium conductances at the mammalian motor nerve terminal*. *Pflugers Arch*, 1987. **410**(4-5): p. 408-12.
87. Schauf, C.L., *Differential sensitivity of amphibian nodal and paranodal K⁺ channels to 4-aminopyridine and TEA*. *Experientia*, 1987. **43**(4): p. 405-8.
88. Kocsis, J.D., et al., *Functional differences between 4-aminopyridine and tetraethylammonium-sensitive potassium channels in myelinated axons*. *Neurosci Lett*, 1987. **75**(2): p. 193-8.
89. Thompson, S., *Aminopyridine block of transient potassium current*. *J Gen Physiol*, 1982. **80**(1): p. 1-18.
90. Lundh, H., S. Leander, and S. Thesleff, *Antagonism of the paralysis produced by botulinum toxin in the rat. The effects of tetraethylammonium, guanidine and 4-aminopyridine*. *J Neurol Sci*, 1977. **32**(1): p. 29-43.
91. Kirsch, G.E. and T. Narahashi, *3,4-diaminopyridine. A potent new potassium channel blocker*. *Biophys J*, 1978. **22**(3): p. 507-12.
92. Kocsis, J.D. and S.G. Waxman, *Long-term regenerated nerve fibres retain sensitivity to potassium channel blocking agents*. *Nature*, 1983. **304**(5927): p. 640-2.
93. Targ, E.F. and J.D. Kocsis, *4-Aminopyridine leads to restoration of conduction in demyelinated rat sciatic nerve*. *Brain Res*, 1985. **328**(2): p. 358-61.
94. Kaji, R. and A.J. Sumner, *Effects of 4-aminopyridine in experimental CNS demyelination*. *Neurology*, 1988. **38**(12): p. 1884-7.
95. Radicheva, N.I. and V.B. Kolev, *4-Aminopyridine and tetraethylammonium-induced changes in action potentials of unmyelinated axons*. *Acta Physiol Pharmacol Bulg*, 1992. **18**(1): p. 21-6.

96. Sobko, A., et al., *Heteromultimeric delayed-rectifier K⁺ channels in schwann cells: developmental expression and role in cell proliferation*. J Neurosci, 1998. **18**(24): p. 10398-408.
97. Fieber, L.A., et al., *Delayed rectifier K currents in NF1 Schwann cells. Pharmacological block inhibits proliferation*. Neurobiol Dis, 2003. **13**(2): p. 136-46.
98. Pappas, C.A. and J.M. Ritchie, *Effect of specific ion channel blockers on cultured Schwann cell proliferation*. Glia, 1998. **22**(2): p. 113-20.
99. Amedee, T., et al., *Voltage-dependent calcium and potassium channels in Schwann cells cultured from dorsal root ganglia of the mouse*. J Physiol, 1991. **441**: p. 35-56.
100. Chandy, K.G., et al., *Voltage-gated potassium channels are required for human T lymphocyte activation*. J Exp Med, 1984. **160**(2): p. 369-85.
101. Sidell, N., et al., *Potassium channels in human NK cells are involved in discrete stages of the killing process*. J Immunol, 1986. **137**(5): p. 1650-8.
102. Schell, S.R., et al., *The inhibitory effects of K⁺ channel-blocking agents on T lymphocyte proliferation and lymphokine production are "nonspecific"*. J Immunol, 1987. **139**(10): p. 3224-30.
103. Sharma, B., *Inhibition of the generation of cytotoxic lymphocytes by potassium ion channel blockers*. Immunology, 1988. **65**(1): p. 101-5.
104. Choquet, D. and H. Korn, *Mechanism of 4-aminopyridine action on voltage-gated potassium channels in lymphocytes*. J Gen Physiol, 1992. **99**(2): p. 217-40.
105. Partiseti, M., et al., *Differential regulation of voltage- and calcium-activated potassium channels in human B lymphocytes*. J Immunol, 1992. **148**(11): p. 3361-8.
106. Cabado, A.G., et al., *Crosstalk between cytosolic pH and intracellular calcium in human lymphocytes: effect of 4-aminopyridin, ammonium chloride and ionomycin*. Cell Signal, 2000. **12**(8): p. 573-81.
107. Barbar, E., et al., *Protein kinase C inhibits the transplasma membrane influx of Ca²⁺ triggered by 4-aminopyridine in Jurkat T lymphocytes*. Biochim Biophys Acta, 2003. **1622**(2): p. 89-98.
108. Lajdova, I., et al., *4-Aminopyridine activates calcium influx through modulation of the pore-forming purinergic receptor in human peripheral blood mononuclear cells*. Can J Physiol Pharmacol, 2004. **82**(1): p. 50-6.
109. Schlichter, L.C. and I.C. MacCoubrey, *Interactive effects of Na and K in killing by human natural killer cells*. Exp Cell Res, 1989. **184**(1): p. 99-108.
110. Hu, D., et al., *Involvement of the 4-aminopyridine-sensitive transient A-type K⁺ current in macrophage-induced neuronal injury*. Eur J Neurosci, 2010. **31**(2): p. 214-22.
111. Lowry, M.A., J.I. Goldberg, and M. Belosevic, *Induction of nitric oxide (NO) synthesis in murine macrophages requires potassium channel activity*. Clin Exp Immunol, 1998. **111**(3): p. 597-603.
112. Qiu, M.R., T.J. Campbell, and S.N. Breit, *A potassium ion channel is involved in cytokine production by activated human macrophages*. Clin Exp Immunol, 2002. **130**(1): p. 67-74.
113. Schilling, T. and C. Eder, *Lysophosphatidylcholine- and MCP-1-induced chemotaxis of monocytes requires potassium channel activity*. Pflugers Arch, 2009. **459**(1): p. 71-7.
114. Hirota, K., et al., *Delivery of rifampicin-PLGA microspheres into alveolar macrophages is promising for treatment of tuberculosis*. J Control Release, 2010. **142**(3): p. 339-46.
115. Ohashi, K., et al., *One-step preparation of rifampicin/poly(lactic-co-glycolic acid) nanoparticle-containing mannitol microspheres using a four-fluid nozzle spray drier for inhalation therapy of tuberculosis*. J Control Release, 2009. **135**(1): p. 19-24.
116. Hirota, K., et al., *Phagostimulatory effect of uptake of PLGA microspheres loaded with rifampicin on alveolar macrophages*. Colloids Surf B Biointerfaces, 2011. **87**(2): p. 293-8.
117. Simon-Yarza, T., et al., *PEGylated-PLGA microparticles containing VEGF for long term drug delivery*. Int J Pharm, 2013. **440**(1): p. 13-8.
118. Makadia, H.K. and S.J. Siegel, *Poly Lactic-co-Glycolic Acid (PLGA) as Biodegradable Controlled Drug Delivery Carrier*. Polymers (Basel), 2011. **3**(3): p. 1377-1397.
119. Terry, A.B., et al., *PLGA nanoparticles for the sustained release of rifampicin*. NGNM, 2014. **2**: p. 1-9.

Please find enclosed the final reports on your manuscript. We are pleased to inform you that your manuscript is accepted for publication and is now being sent to our publisher to be included in the

next available issue of EMBO Molecular Medicine.

We would like to remind you that as part of the EMBO Publications transparent editorial process initiative, EMBO Molecular Medicine will publish a Review Process File online to accompany accepted manuscripts. If you do NOT want the file to be published or would like to exclude figures, please immediately inform the editorial office via e-mail.

If you want to receive an e-mail alert regarding its publication as well as other EMBO Mol Med content, register here: <http://embomolmed.embopress.org/alerts>

Our RSS feeds can be found at feed:
<http://embomolmed.embopress.org/rss>

Please read below for additional IMPORTANT information regarding your article, its publication and the production process.

Congratulations on your interesting work,

***** Reviewer's comments *****

Referee #1 (Comments on Novelty/Model System):

The authors have gone to great lengths to address the reviewers comments and thus have significantly improved the manuscript.

Referee #1 (Remarks):

Well done, a potentially very important study.

Referee #2 (Comments on Novelty/Model System):

Prof Noble has made a strong revision and rebuttal and I am happy with these changes.

Referee #2 (Remarks):

Prof Noble has made a detailed, intelligent, eloquent rebuttal with additions to the manuscript. I feel the manuscript is now suitable for publication.

Corresponding Author Name: John Elfar

Manuscript Number: EMM-2015-06035-V2